

# Nocturnal boundary layer turbulence regimes analysis during the BLLAST campaign

Jesús Yus-Díez[1,5], Mireia Udina[1], Maria Rosa Soler[1], Marie Lothon[2], Erik Nilsson[3], Joan Bech[1], and Jielun Sun[4]

[1]Departament de Física Aplicada - Meteorologia, Universitat de Barcelona, C/Martí i Franquès, 1., 08028, Barcelona, Spain
[2]Laboratoire d'Aérologie, University of Toulouse, CNRS, France
[3]Department of Earth Sciences, Uppsala University, Uppsala, Sweden
[4]NorthWest Research Associates, Boulder, Colorado, USA
[5]Institute of Environmental Assessment and Water Research (IDAEA-CSIC), C/Jordi Girona 18-26, 08034, Barcelona, Spain
**Correspondence:** Jesús Yus-Díez (jesus.yus@idaea.csic.es), Mireia Udina (mudina@meteo.ub.edu)

**Abstract.** A night-time turbulence regime classification, the so-called HOckey-Stick Transition (HOST) theory, proposed by Sun et al. (J Atmos Sci 69(1):338-351, 2012) from the Cooperative Atmosphere-Surface Exchange Study 1999 (CASES-99) is explored using data from the Boundary-Layer Late Afternoon and Sunset Turbulence (BLLAST) field campaign which took place during the summer 2011 in the north of the central French Pyrenean foothills.

Results show that the HOST turbulence relationships for the BLLAST field campaign data is strongly dependent on both the meteorological and orographic features. The HOST pattern only appears for nights when a stably stratified boundary layer can be developed, corresponding to fair weather and clear sky nights, when the flow is generated by the nearby orography, from the south and southeast directions. Those flows strongly influenced by the orography may generate intermittent or enhanced turbulence. When considering the whole dataset for these flow directions, several enhanced turbulence points are found to be

associated with sudden wind speed and directional shear transitions. In contrast, flows from other directions do not reproduce the HOST relationship and the turbulence relationship is almost linear, independent of the vertical temperature gradients, corresponding to flows driven by mesoscale or synoptic scales. In addition we identify examples of gravity waves and top-down turbulent events that lead to transitions between the turbulence regimes.

**1    Introduction**

With the aim to investigate and categorise turbulence patterns generated by wind shear in the atmospheric stable boundary layer (SBL), Sun et al. (2012) analysed a month-long dataset collected in Kansas from the Cooperative Atmosphere-Surface Exchange Study in October 1999 (CASES-99) (Poulos et al., 2002). They found that, depending on the relationship between the turbulence intensity of the flow and the wind velocity, turbulent mixing from the surface to the last level of measurement





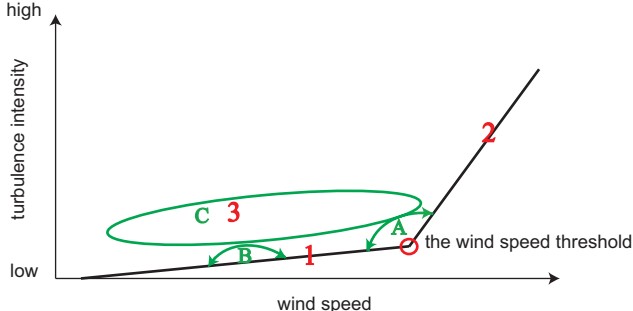

**Figure 1.** Schematic representation of the HOckey-STick (HOST) theory with the three turbulence regimes (regime 1, 2 and 3) and the three turbulence intermittency categories (cat. A, cat. B and cat. C). Reproduced from Figure 2 of Sun et al. (2012) ©American Meteorological Society. Used with permission.

(55m) can be categorised in three regimes (Fig. 1). A regime 1 occurring when the wind speed ($V$) is less than a threshold value ($V_T$), characterised by low turbulence intensity increasing slightly with the velocity. This is the weak turbulence regime, which is generated by local shear and modulated by the vertical temperature gradients of the SBL. Eddies generated by local shear, $\delta V/\delta z$, do not interact with the ground as their length scale is smaller than the observation height z, $\delta z < z$ (Sun et al., 2012). When $V > V_T$ regime 2 occurs and turbulence is mainly driven by the bulk shear, therefore increasing strongly and nearly linearly with V. The close relationship between turbulence intensity and mean wind speed suggests that near the ground, turbulence under moderate winds responds to the bulk shear $V(z)/z$ (Sun et al., 2012, 2016). The eddies dominating this stronger turbulence regime have a larger scale of $z$, thus leading to a well mixed layer below this height and producing a near-neutral stratification. The value of the wind speed threshold at which the shear changes from local to bulk increases with height approximately logarithmically (Sun et al., 2012; Bonin et al., 2015). In addition to these two turbulent regimes generated by the local or the bulk shear, moderate turbulence can be often generated for relative low values of V above the observation level by different coherent structures, for instance: Kelvin- Helmholtz instabilities (Newsom and Banta, 2003), gravity waves (Sun et al., 2004; Udina et al., 2013), and density currents (Sun et al., 2002; Viana et al., 2010; Soler et al., 2014) which can generate turbulence, that is transported downward, toward observation levels where the wind speed is smaller than the threshold value. This top-down turbulent regime is defined as regime 3, which also includes turbulence enhanced by wind shear associated to topographical features leading to flow accelerations, directional wind shear and sub-meso motions (Mahrt et al., 2013). All these intermittent turbulent events have in common the non-stationarity of the flow.

The schematic representation of the transition between the weak and the strong turbulence regimes resembles a HOckey-Stick (Fig. 1), this is why this theory is named Hockey-stick Transition (HOST). It can be explained in terms of total turbulence energy (TTE) defined as the sum of the turbulent kinetic energy (TKE) and the turbulent potential energy (TPE), also called available potential energy. As shown in Sun et al. (2016), if the vertical variation of TTE is small, the turbulence shear production controls the variation of TTE. When $V < V_T$ the vertical transport of cold air is confined near the surface, the stable



stratification of the flow increases. Therefore more turbulence energy is used to increase the TPE (Lin and Pao, 1979), and TKE does not increase significantly with increasing $V(z)$. This situation corresponds to regime 1. However, if $V > V_T$ the stable stratification is considerably reduced, leading to a near-neutral regime. As a consequence, shear generates TKE as the TTE is not consumed to increase TPE, this situation corresponds to regime 2.

The HOST theory represents a new approach for the turbulence parameterisation in the surface layer, as the Monin Obukhov Similarity Theory (MOST) may be limited, because the bulk formulae are applicable only within a near thin surface layer (Sun et al., 2012, 2016). The theory has been previously studied for other campaigns and sites in relatively flat terrain areas. Mahrt et al. (2013) studied the turbulence behaviour in three sites with different surface roughness, showing that the wind speed threshold decreases with increasing roughness length; Andreae et al. (2015) explored the three turbulence regimes and

characterised the nocturnal boundary layer using data from The Amazon Tall Tower Observatory (ATTO); Bonin et al. (2015) used the HOST via remote sensing in the Southern Great Plains and Acevedo et al. (2016) studied the HOST theory via the contrasting structures of the stable and neutral states of the boundary layer over snow-covered surfaces. The theory has also been investigated through large eddy simulation (LES) models (Udina et al., 2016), revealing the LES difficulties to reproduce regime 1.

As the HOST was originally derived from measurements taken in relatively flat area, where the CASES-99 field campaign was held (Sun et al., 2012), the aim of this study is to explore the validity of the HOST theory in a complex terrain area, for flows that are influenced by the nearby orography. The data analysed herein is from the Boundary Layer Late Afternoon and Sunset Turbulence (BLLAST) campaign (Lothon et al., 2014), which took place during early summer of 2011 in France, in the "Plateau de Lannemezan", north of the central French Pyrenean foothills.

The structure of the paper is as follows. The site, the in situ observations of near surface turbulence obtained from the array of instruments and the data processing method applied are described in Sect. 2. The wind flow and wind regimes affecting the site during the BLLAST field experiment are analysed in Sect. 3. The categorisation of the turbulence regimes patterns for the BLLAST field campaign is investigated in sections 4 and 5, while the transition between regimes is illustrated in Sect. 6 using selected examples. A summary and the main conclusions are given in Sect. 7.

## 2   Site, observations and data processing

Data used in this study comes from the BLLAST field campaign conducted in early summer, from 14 June to 8 July 2011 in France around the Centre for Atmospheric Research (CRA, *Centre de Recherches Atmosphériques*) (Lothon et al., 2014). The site (site 1 in Fig. 2a) is located in the region called "Plateau de Lannemezan", which is a plateau over the Garonne basin at about 600 m above sea level (a.s.l.), a few kilometres north to the Pyrenean foothill, with pikes of around 1500-2000 m

a.s.l., and about 45 km away from the highest peaks of the Pyrenees mountain range, at around 3000 m a.s.l. in the border with Aragón, Spain. The site is influenced by the steep orography at the southwest (Pic du midi de Bigorre) and the Aure and Garonne valleys located at the south and east-southeast, respectively (Fig. 2a).



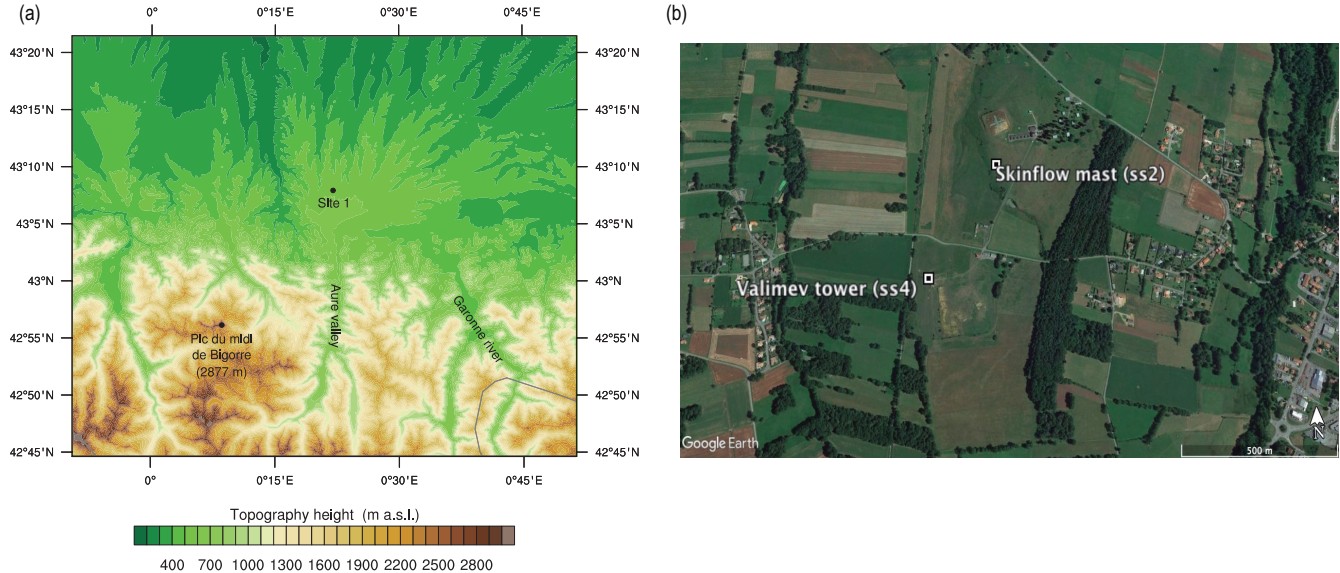

**Figure 2.** Location of the "Plateau de Lannemezan" and the surrounding area with: (a) a topographical map of the surroundings of site 1 where the BLLAST field campaign took place showing terrain elevation of the area, with the Pyrenees mountain range at its south and the location of the Aure valley and the Garonne river, and (b) aerial view zoom to the site 1 with the location of the Skinflow mast at surface-site 2 (ss2) and the Valimev tower at surface-site 4 (ss4).

The main objective of the BLLAST field campaign was to improve the knowledge of the afternoon and evening transition in the Boundary Layer (BL) (Lothon et al., 2014). To achieve this, a dense array of instrumentation was deployed over the area, although only some of them are employed herein (Table 1). We focus the analysis on the data from the 10 m mast of the surface-site 2 (ss2), called Skinflow mast, and the 60 m tower placed at the surface-site 4 (ss4), called Valimev tower (Fig. 2b).

5   From the Skinflow mast we use data from the sensors located at 2.23 m, 3.23 m, 5.27 m and 8.22 m (named hereafter $z_{2m}$, $z_{3m}$, $z_{5m}$ and $z_{8m}$, respectively) and from the Valimev tower we use data from the sensors located at 29.4 m, 45.8 m and 61.4 m (named hereafter $z_{30m}$, $z_{45m}$ and $z_{60m}$, respectively), see Table 1. Since the analysis is restricted to the night-time, we only consider data measured from 1942 UTC to 0420 UTC, the sunset and sunrise hours on 27 June 2011, which is the day in the middle of the study period.

10   From the Skinflow mast thermocouples and sonic anemometers deployed between $z_{2m}$ and $z_{8m}$ we have used two different sets of data. The $z_{2m}$ level is omitted when results were very similar to those from $z_{3m}$. The timespan of available measurements for the Skinflow mast levels starts on the 19 June 2011 and finishes on the 8 July 2011. Firstly, a set with a temporal average of 5 min is used for a statistical study of the behaviour of the turbulence strength relationship with the wind speed (Sun et al., 2012; Mahrt, 2014; Sun, 2011), where the high frequency signal of the turbulence is used to calculate average statistics to show a broad picture of the patterns. Secondly, a small temporal average set with averages ranging between 10 s and 1 min is



**Table 1.** Skinflow mast and Valimev tower sonic measurements from the BLLAST field campaign used in this study.

| Level name | Height [m] | Measured variables | Sensors | Data period |
|---|---|---|---|---|
| $z_{2m}$ | 2.23 | | | |
| $z_{3m}$ | 3.23 | | Campbell Thermocouple E-Type FW05 | 19 June 2011 - 8 July 2011 |
| $z_{5m}$ | 5.27 | $T$ [°C], $(u,v,w)$ [m s$^{-1}$] | Campbell Csat3 3D sonic anemometer | |
| $z_{8m}$ | 8.22 | | | |
| $z_{30m}$ | 29.4 | | Campbell Csat3 3D sonic anemometer | 14 June 2011 - 8 July 2011 |
| $z_{45m}$ | 45.8 | | Gill master pro 3D sonic anemometer, wind vane | |
| $z_{60m}$ | 61.4 | | Campbell Csat3 3D sonic anemometer | 15 June 2011 - 8 July 2011 |

obtained. This high-frequency data aims to show with more detail the temporal evolution of some variables which account for the turbulence intermittency within and between the different turbulent regimes defined in the HOST theory.

From the Valimev 60 m tower we have used the measurements from the sonic anemometers located at the $z_{30m}$, $z_{45m}$ and $z_{60m}$ levels, as shown in Table 1. The dataset for this tower starts on the 14 June of 2011 for the $z_{30m}$ and $z_{45m}$ levels, and

slightly later for $z_{60m}$, between 1000-1100 UTC of the 15 June 2011, and finishes on the 8 July 2011 for the three levels. The same data processing as to the Skinflow mast levels measurements is applied to the Valimev levels, using temporal averages of 5 min and averages ranging between 10 s and 1 min as well.

We apply the following corrections and filters, according to Said et al. (2011a, b). The wind direction at the $z_{30m}$, $z_{45m}$ and $z_{60m}$ levels is modified by adding a constant phase $\phi = -90°$ to account for the orientation of the sonic anemometer.

Moreover, wind directions computed from the horizontal wind components, $u$ and $v$, need phase corrections of 318.71° for the sonic anemometer at $z_{30m}$, $-73.8°$ for the Gill sonic anemometer at $z_{45m}$ $-21.5°$ for the wind vane at $z_{45m}$ and 329.71° for the sonic anemometer at $z_{60m}$. Additionally, wind components were discarded when the difference between the ultrasonic estimation at $z_{30m}$, $z_{45m}$, and $z_{60m}$ and the wind vane at $z_{45m}$ were greater than a threshold of 2.5 m s$^{-1}$, 1.5 m s$^{-1}$, and 3.5 m s$^{-1}$, respectively. The standard deviation for the horizontal and vertical velocity greater than 4 m s$^{-1}$ and 2 m s$^{-1}$,

respectively, are also discarded. Finally, sonic measurements were not considered when anomalous TKE values were observed during heavy rain periods.

We calculate several parameters in order to study the turbulence regimes relationships: the wind speed, $V$:

$$V = \sqrt{u^2 + v^2} \tag{1}$$

where $u$ and $v$ are the horizontal wind components; the turbulence intensity, $V_{TKE}$:

$$V_{TKE} = \sqrt{\frac{1}{2}(\sigma_V^2 + \sigma_w^2)} \tag{2}$$





where $\sigma$ represents the standard deviation of the wind components, and $\sigma_V^2 = \sigma_u^2 + \sigma_v^2$; the potential virtual temperature:

$$\theta_v = T_v + \Gamma_d z \qquad (3)$$

where $T_v$ is the air virtual temperature at each height $z$ and $\Gamma_d$ is the dry adiabatic lapse rate; and the skewness of the vertical velocity fluctuations (Wyngaard, 2010):

$$S_w = \frac{\overline{w'(z)^3}}{\left(\overline{w'(z)^2}\right)^{3/2}} \qquad (4)$$

which gives us information about the vertical transport (upward or downward) of the vertical velocity variance.

Furthermore, when the relationship $V_{TKE}$ vs V is graphically analysed we use the bin-average technique where we group the data into wind speed bins of $0.4 \ \mathrm{m \ s^{-1}}$ starting at a wind speed of $0.1 \ \mathrm{m \ s^{-1}}$. For each of those bins it is displayed the median value with the corresponding error bars. These error bars represent the first and third quartiles instead of the actual standard deviation as a consequence of the skewed distribution of the turbulence points for each wind speed bin. This bin-averaged technique is also employed for the box plot analysis. In addition, we have made an exhaustive analysis in order not to consider the measurements collected under strong precipitation events, where the sonic anemometers do not work properly.

## 3 Flow characterisation during the BLLAST field campaign

The objective of the paper is to study how the HOST theory behaves in the complex terrain environment for the BLLAST field experiment in the nocturnal boundary layer. In the area, during night-time when thermally-driven flows can be developed, a katabatic flow going down from the mountain to the plain is established, through valley and downslope winds (Román-Cascón et al., 2018). Since the HOST theory proposed by Sun et al. (2012, 2016) during the CASES-99 campaign corresponds to nights with clear skies and intense radiative cooling allowing the development of local and mesoscale winds, we first select the nights that meet these conditions during the BLLAST campaign. This is done similarly as Lothon et al. (2014) selected the intensive observation periods (IOPs) in BLLAST, focused on the diurnal and the late-afternoon transition boundary layer analysis. Therefore, we consider as IOP nights, hereafter nIOPs, the period between 1942 UTC of that IOP day and 0420 UTC of the following day (Blay-Carreras et al., 2014). That is, the evening, night and early morning following an IOP day, excluding any nocturnal period strongly influenced by synoptic phenomena, such as frontal system or mesoscale convective systems. In the case when the night-time period prior to the IOP shows the typical nocturnal flow cycle, we do consider the early morning of the corresponding IOP, i.e. from 0000 to 0420 UTC. Following these criteria, table 2 shows the nocturnal periods considered as nIOPs.

To characterise the wind flow regimes on the Plateau de Lannemezan for the nIOPs, the first step is to calculate the wind roses for each level of measurement, both for the Skinflow mast and the Valimev tower. Fig. 3 shows that most of the flows ($> 60 \ \%$) at the lower levels come from the southeast quadrant, corresponding to shallow drainage flows formed after sunset due to local small slopes located in the foothills of the Pyrenees (Román-Cascón et al., 2015). At the higher levels wind roses show a large fraction of winds coming from the southeast quadrant ($40 \ \%$ of the data), associated with the larger scale mountain



**Table 2.** Night-time data, between 1942 and 0420 UTC, of the Skinflow mast and Valimev tower levels considered as IOPs nights. The temporal series starts the 14 of June for the first two levels with sonic anemometers of the Valimev tower, and the 15 of June for $z_{60m}$; for the Skinflow data, the measurements began the 19 of June.

| nIOP | Corresponding night-time IOP | Date (dd/mm) | Levels |
|---|---|---|---|
| nIOP00 | IOP00-IOP01 | 14/06-15/06 | $z_{30m}, z_{45m}$ |
| nIOP01 | IOP02-IOP03 | 19/06-20/06 | $z_{30m} - z_{60m}$ |
| nIOP02 | IOP03 | 20/06 (1942-2359 UTC) | $z_{3m} - z_{60m}$ |
| nIOP03 | IOP04-IOP05 | 24/06-25/06 | $z_{3m} - z_{60m}$ |
| nIOP04 | IOP05-IOP06 | 25/06-26/06 | $z_{3m} - z_{60m}$ |
| nIOP05 | IOP06-IOP07 | 26/06-27/06 | $z_{3m} - z_{60m}$ |
| nIOP06 | IOP07 | 27/06 (1942-2100 UTC) | $z_{3m} - z_{60m}$ |
| nIOP07 | IOP08-IOP09 | 30/06-01/07 | $z_{3m} - z_{60m}$ |
| nIOP08 | IOP09-IOP10 | 01/07-02/07 | $z_{3m} - z_{60m}$ |
| nIOP09 | IOP10 - 03/07 | 02/07-03/07 | $z_{3m} - z_{60m}$ |
| nIOP10 | IOP11 | 05/07 (0000 - 0420 UTC) | $z_{3m} - z_{60m}$ |

plain circulation (Román-Cascón et al., 2015), with another large fraction coming from the northwest quadrant ($25 - 30$ % of all data) associated to the typical synoptic flows affecting the site (Barneoud et al., 2010). In BLLAST it is a common feature during the IOPs the presence of diurnal anabatic winds from the valleys located north of the site, which progressively veer towards the south as the sun sets, firstly nearby the mountains and then as katabatic winds flowing through the valley towards

the site (Jimenez and Cuxart, 2014). The occurrence of the north-westerlies at high levels could be associated with the fact that the SBL top may be below the higher measurements levels of the Valimev tower, thus the synoptic forcing is too strong, and the wind is forced by this scale. Another relevant detail that has to be addressed and that is perceived in the wind roses of the low levels is that the Skinflow mast is located at the north part of a very small gully (Fig. 2b), a formation of a very local shallow drainage flow is possible (Nauta, 2013; Soler et al., 2002) during the very SBL.

An example of these shallow drainage flows (SDF) and mountain plain flows (MPF) is shown in Román-Cascón et al. (2015), between noon of the 1st of July and the morning of the 2nd July 2011, including nIOP08. The predominant direction during the central part of the day-time is from the north corresponding to the uphill direction (Jimenez and Cuxart, 2014), with a wind speed on average of $2 \text{ m s}^{-1}$. Later that day, the 1st of July, at 1800 UTC, the wind turns to south-southeast, first at the lower levels, which correspond to drainage winds with lower velocities, and two hours later, at 2000 UTC, the wind turns to southeast

at all the levels increasing its velocity up to $5 \text{ m s}^{-1}$ associated to a mountain plain flow. The night flow for those stable nights has an intermittent nature since it is generated by drainage flows, mountain-plain and other mesoscale phenomena induced by the temperature gradients between the plains and the most septentrional mountain ranges of the Pyrenees located at the south of the site.





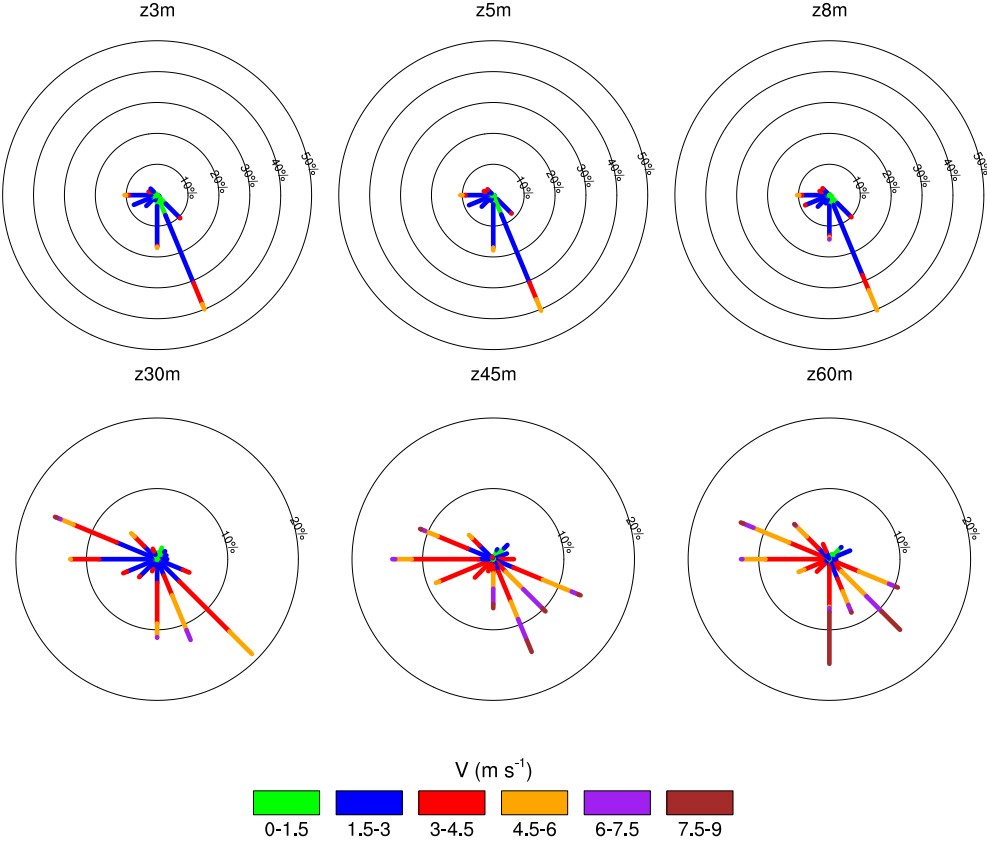

**Figure 3.** Night-time IOPs (nIOPs) wind roses for each level of the Skinflow mast and the Valimev tower during the BLLAST field campaign. Circles represent a 10% frequency increment.

Even though the main focus is the study of the HOST theory for the SBL, we also aim to study the behaviour of such theory for non nIOP conditions, and how the peculiarities of the Plateau de Lannemezan site affect its structure. Bearing this in mind we represent the wind rose for all the night-time set of data from the same levels of both towers (Fig. 4).

Winds from the west and west-northwest direction are very frequent in this part of Europe area, with additional channelizing
5    to westerlies due to the mountain range at the south of the site. A frequency of 45% is found for the Valimev tower dataset and 35% for the Skinflow mast. Its occurrence is mainly linked to larger-scale synoptic and mesoscale conditions (Barneoud et al., 2010). Southeast winds account for about 25% and 35% of the flow for the Valimev and the Skinflow mast, respectively. Some of this SE flows are associated to SDFs (Jimenez and Cuxart, 2014), that are measured at the Skinflow mast and are





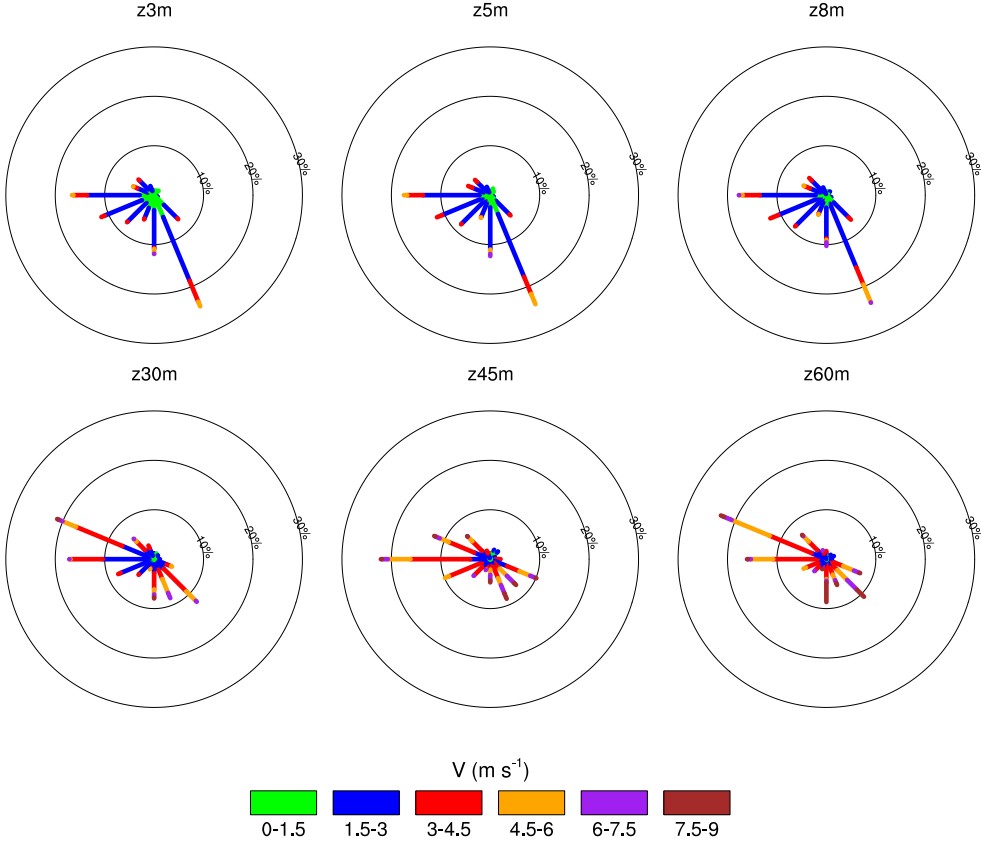

**Figure 4.** Night-time whole dataset wind roses for each level of the Skinflow mast and the Valimev tower during the BLLAST field campaign. Circles represent a 10% frequency increment.

typically eroded by the arrival of the deeper and larger scale MPF, measured at all the levels with an increase in wind speed and veering towards the southwest. This SW wind is largely associated to the scale of the Pyrenees, the MPF, and represents 20% of the flow for the lower levels and 15% for the higher ones. The difference between the higher and the lower levels wind directions distribution, Fig. 4, can be associated with the influence of the meteorological phenomena at different levels, the shallow drainage flows, mountain plain winds, and synoptic scale flows.

If we compare results from Fig. 3 and Fig. 4 we can clearly see the larger proportion of SE flows arriving to the site when the nIOPs are considered, whereas when the wind rose is referred to the whole BLLAST dataset, other directions, as W-



NW increase its relevance. Indeed, this behaviour is even more pronounced for the Skinflow mast levels, where the western component for the nIOPs dataset is almost non-existent or very scarce.

After having classified and illustrated the flow circulation from the dataset, we analyse the turbulence regimes from two different perspectives: using only nIOPS data (Sect. 4) and using the whole BLLAST dataset (Sect. 5).

## 4   Turbulence regimes analysis: nIOPs

In this section we investigate the turbulence regimes using the data obtained from the so-called nIOPs in order to study the generality of the HOST pattern (Sun et al., 2012). We explore the dependence of the night-time turbulence intensity, $V_{TKE}$, with the horizontal wind speed $V$ for each of the observational levels. In addition, to deeper illustrate this relationship and its variability we use box plots over the measured levels. Finally we analyse the $V_{TKE}$ $vs$ $V$ relationship dependence on the thermal stratification through bulk potential temperature differences between the Skinflow mast levels.

### 4.1   Turbulence relationship

Fig. 5a shows the relationship between $V_{TKE}$ and V, where the wind speed threshold is marked for every level (see triangles in Fig. 5a). The turbulence intensity relationships with $V$ for the nIOPs have a similar hockey stick shape as the one obtained by Sun et al. (2012, 2016) using data from CASES-99. As described schematically in Fig. 1, Fig. 5a shows that for a given height, below a wind speed threshold, $V_T$, the turbulence intensity does not depend on the wind speed with no actual increase of $V_{TKE}$ (regime 1). On the contrary, above a given $V_T$, $V_{TKE}$ increases much more strongly than for lower wind speeds (regime 2). Table 3 shows the values of the wind speed thresholds obtained for the BLLAST field campaign during nIOPs. As observed by Sun et al. (2012) and Bonin et al. (2015), the wind speed threshold, $V_T$, (table 3) also increases logarithmically with height in the BLLAST site during the nIOPs and the $r^2$ correlation between the logarithmic height and the wind speed is around 0.92. The threshold value marks a change in the slope of the turbulence intensity relationship. Therefore, we consider that the regime changes to regime 2, when the slope has increased at least by two times the value of the slope of regime 1. The higher the layer, i.e. the deeper the stratus below z, the greater the value of $V$ required for the layer to become neutral. Therefore, the $V_T$ increases with height because larger shear production is needed to compensate for inhibiting effect that near-surface stable stratification has on TKE. The $V_T$ values for nIOPs in BLLAST are lower than those obtained by Sun et al. (2012) for CASES-99 and the bulk shear ($V_T/z$) is weaker as well (Fig. 5b). This can be explained by the effect of a larger roughness length in the BLLAST site, associated to the mountainous terrain south of the site. The greater the roughness length, the smaller the $V_T$, which means that weaker wind is necessary to achieve regime 2, as also seen by Mahrt et al. (2013). It is also noticeable that the slope for the regime 2 for all the levels is quite similar, which indicates that, as expected, once the layer is neutral, the relationship between the turbulence intensity and the wind speed tends to be independent of the layer depth. Although the three turbulent regimes of the HOST pattern are visible in all levels, in the higher level, $z_{60m}$, the regime 2 does not behave as expected for the greatest values of the wind speed, showing a decrease in the turbulence intensity as the $V$ increases. After comparing the evolution of the main magnitudes at different levels, a possible explanation is that the higher




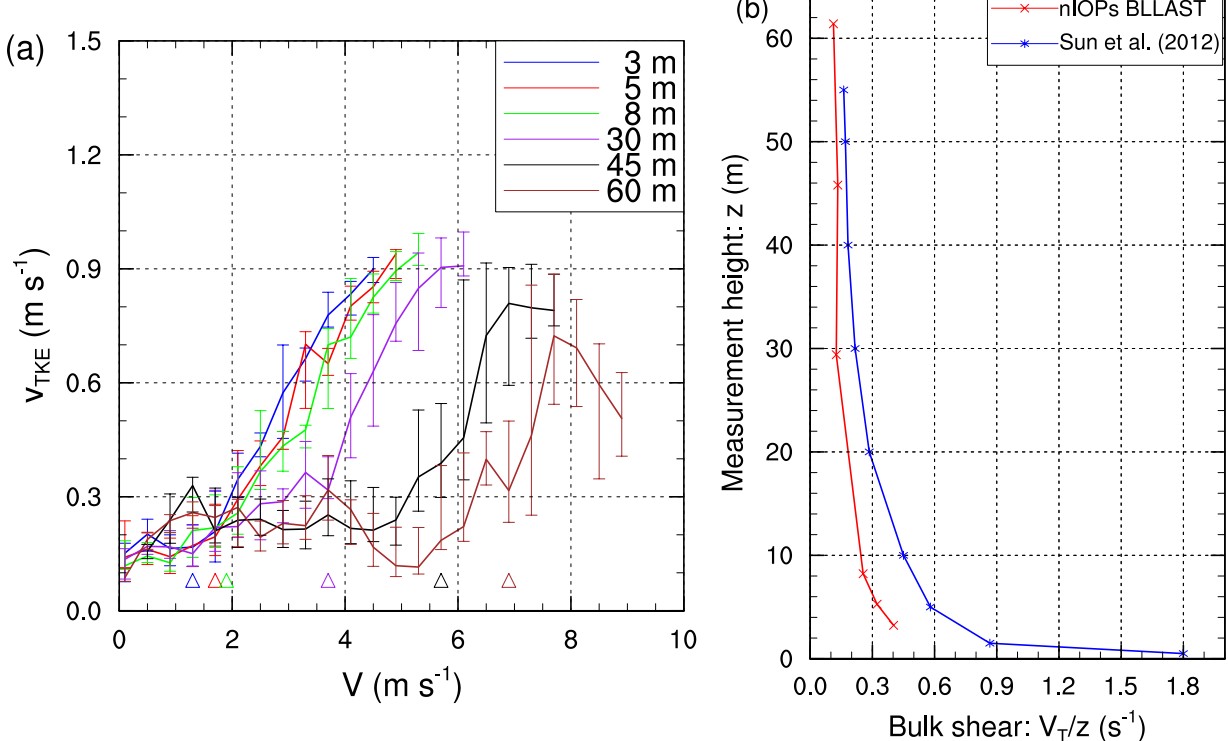

**Figure 5.** (a) Relationship between the bin-averaged turbulence intensity $V_{TKE}$ and the wind speed $V$ for the nIOPs of the BLLAST campaign. The lines join the median value of each $V$ bin ($0.4\,\mathrm{m\,s^{-1}}$ bins) for each level of measurement. The vertical lines of each bin mark the first and third quartiles with lower and upper end, respectively. The wind speed threshold $V_T$ is marked with a coloured triangle for each coloured level. (b) Bulk shear variation with height obtained with BLLAST nIOPs data (red line) compared with CASES99 data reported by Sun et al. 2012 (blue line).

**Table 3.** Wind speed thresholds $V_T$ for the Skinflow mast and Valimev tower levels for the BLLAST field campaign.

|  | $z_{3m}$ | $z_{5m}$ | $z_{8m}$ | $z_{30m}$ | $z_{45m}$ | $z_{60m}$ |
|---|---|---|---|---|---|---|
| $V_T$ (m s$^{-1}$) | 1.3 | 1.7 | 2.1 | 3.7 | 6.1 | 6.9 |

level is often decoupled from the layers below, and could possibly be located above the top of the SBL. However, the relatively small number of points at these velocities may not be enough to get representative results.

The relationships of $\sigma_V$ and $\sigma_W$ with $V$ (not shown) also present the HOST dependence seen for $V_{TKE}$. However, the turbulence values are generally smaller for $\sigma_W$ with a smaller increase of the vertical variance with increasing V in regime 2 due to the ground impingement as found in Sun et al. (2012).



## 4.2 Box plots analysis

In order to further understand the turbulence behaviour during the nIOPs, we illustrate the relationship between $V_{TKE}$ and $V$ using the box plots for each of the measured levels. This technique is only useful if the number of points for each of the boxes is large enough, otherwise the result may be senseless, therefore we only represent the box if the number of measurements

falling within the bin is more than a minimum of 5. The method is a simple form to represent several simple statistics, such as the minimum and the maximum, the lower and upper quartiles and the median in a visual display.

Each box for each bin contains the data between the first (Q1) and the third quartile (Q3), called interquartile range (IQR), with the median within. The marks representing the values of $Q1 - 1.5 \cdot IQR$ and $Q3 + 1.5 \cdot IQR$ correspond to the lower and the upper whiskers respectively, unless the maximum (minimum) value is lower (higher) than those values. The outliers are

those points falling outside of the whiskers. The amplitude of the box, i.e. the IQR, depends on the dispersion of the points within the bin, if $V_{TKE}$ varies considerably for a given value of V, then the box will be larger. The whiskers represent how far the dispersion of $V_{TKE}$ values are from the median. The point distribution is not Gaussian for each bin and the whiskers reach further values above than below the median. This skewness is due to the non-existence of negative values and the fact that the turbulence intensity can be enhanced due to the presence of coherent structures. The quantity of outliers indicates the

presence of turbulent events that lead to an increase in the turbulent intensity that differs greatly from the value expected for a given range of V. This situation corresponds to regime 3 characterised by intermittent turbulence phenomena, which is typical for clear and calm nights.

Figure 6 shows that most of the boxes have an IQR relatively small for the regime 1, whilst for the regime 2 the dispersion increases, this is appreciated even more for the levels between $z_{8m}$ and $z_{60m}$ (Fig. 6c, 6d, 6e, 6f). Not only the IQR shows the

dispersion of the points for a certain bin, but the whiskers, as well as the outliers, provide additional information. The number of outliers within regime 3 is larger for wind speeds below the threshold, since the variability of the turbulence is higher. Instead, the strong wind associated to neutral stratification tends to suppress the occurrence of regime 3. It can be noted that for wind speeds closer to the threshold the scatter increases (Fig. 6), which is highly due to the bin-averaged technique and the fact that around this velocity the turbulence intensity increases quicker. The increase in scattering is observed in the IQR values, i.e. the

larger length of the boxes, and in the greater amount of outliers.

There is a big difference in Fig. 6 in the shape and in the characteristics of the box plots corresponding to the Skinflow mast and the Valimev tower. Fig. 6a, 6b and 6c show the same pattern, with very similar IQR for $V < V_T$, where the scattering increases for the bins around the wind speed threshold. In addition, the scattering also increases with a greater number of outliers at higher levels (Fig. 6e and 6f).

## 4.3 Turbulence regimes and thermal stratification


As turbulent mixing is influenced by the vertical temperature gradients, we now consider the effect of stratification on the turbulence relationship. We only proceed for the Skinflow mast because the separation of the measurements layers in the Valimev tower are too deep to calculate the bulk potential temperature difference, $\Delta \bar{\theta}_v$. Herein $\Delta \bar{\theta}_v$ is defined as $\Delta \bar{\theta}_v =$



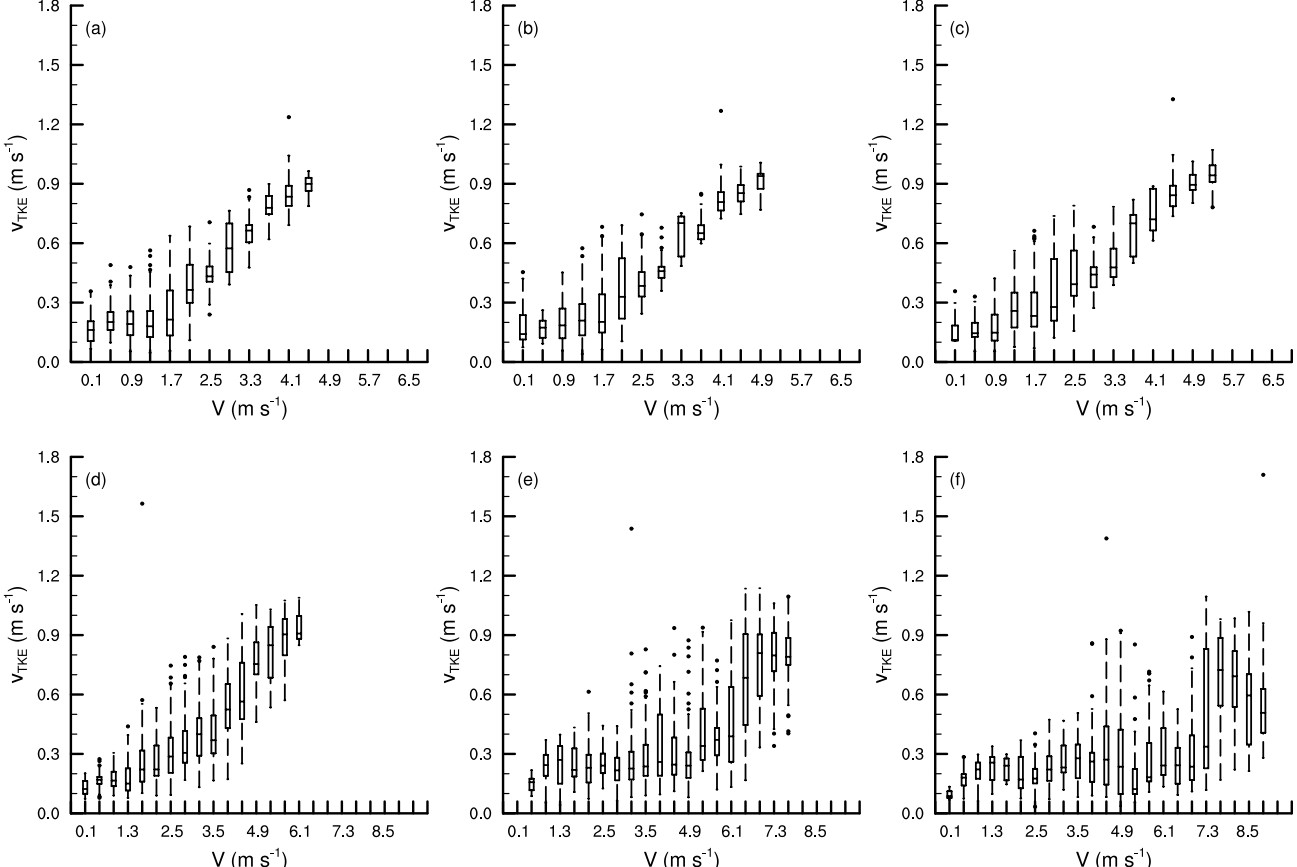

**Figure 6.** Box plots of the relationship between the bin-averaged turbulence strength $V_{TKE}$ and the wind speed $V$ for the nIOPs of the BLLAST field campaign. Data correspond to the Skinflow mast and to the Valimev tower at different heights, (a) $z_{3m}$, (b) $z_{5m}$ and (c) $z_{8m}$ Skinflow mast levels, and (d) $z_{30m}$, (e) $z_{45m}$ and (f) $z_{60m}$ Valimev tower levels.

$\bar{\theta}_v(z) - \bar{\theta}_{v_{z_0}}$, where $\bar{\theta}_{v_{z_0}}$ is the virtual potential temperature of reference at $z_{2m}$, and $\bar{\theta}_v(z)$ the virtual potential temperature at each height z.

In Fig. 7 we observe that for wind speeds greater than the threshold, $V > V_T$, the shear is strong enough to generate eddies with scale z, leading to a boundary layer neutrally stratified with a potential vertical temperature difference close to zero (green line). At all levels (Fig. 7 a, b, c) for wind speeds below the threshold, the stratification of the night-time boundary layer becomes stable, so there is a shallow mixing layer with a depth determined by $\delta z < z$ (Sun et al., 2012). Therefore turbulent eddies do not reach the surface and cold air is accumulated near the surface, resulting in layers with values of $\Delta \bar{\theta}_v > 0.5$ K. However, in Fig. 7 we observe situations that deviate from those described above. For example, in regime 1, for very stable situations ($\Delta \bar{\theta}_v > 0.5$ K), intermittent turbulence between regimes 1 and 2, and between regimes 1 and 3, can occur. Note that





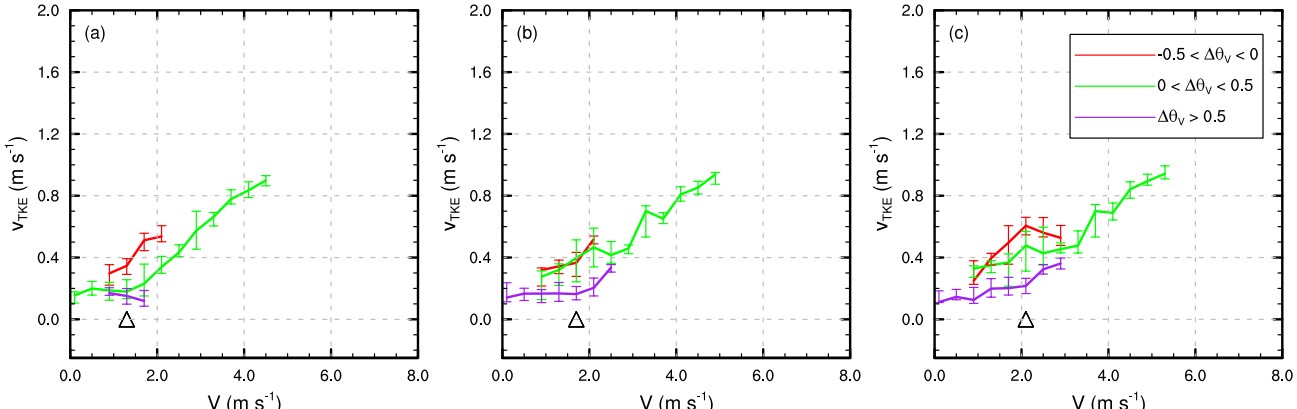

**Figure 7.** Relationship between the bin-averaged turbulence strength $V_{TKE}$ and the wind speed $V$ during the BLLAST campaign nIOPs as a function of potential temperature difference intervals, defined as, $\Delta\bar{\theta}_v = \bar{\theta}_v(z) - \bar{\theta}_{v_{z_0}}$, where $\bar{\theta}_{v_{z_0}}$ is the virtual potential temperature of reference at 2 m, and $\bar{\theta}_v(z)$ the virtual potential temperature at each height z: (a) $z_{3m}$, (b) $z_{5m}$, and (c) $z_{8m}$. The wind speed threshold $V_T$ is marked with a black triangle for each height (a) $z_{3m}$, (b) $z_{5m}$, and (c) $z_{8m}$.

these transitions increase the turbulence, giving rise to temperature variations and mixing that would result in a near neutral or even unstable regime (see green and red lines in Fig. 7).

For $V > V_T$ the turbulent mixing processes tend to decrease the variability of $\Delta\bar{\theta}_v$ (Sun et al., 2016) and the stratification becomes near neutral. As it is expected, for $V < V_T$, the magnitude of the turbulence intensity, $V_{TKE}$, increases with decreasing
values of $\Delta\bar{\theta}_v$, as the relevance of TPE decreases with instability (Sun et al., 2016; Zilitinkevich et al., 2007). Another important point to consider is the thickness of the layer, as $\Delta\bar{\theta}_v$ represents a vertically integrated virtual potential temperature difference, thus it will increase when the height of the level increases (Fig. 7 a, b, c). Therefore, due to the proximity between the $z_{3m}$ and $z_{2m}$ levels most of the points fall into the stable-neutral stratification $0\,\mathrm{K} < \Delta\bar{\theta}_v < 0.5\,\mathrm{K}$, which explains the predominance of the green line in Fig. 7a. For the $z_{5m}$ and $z_{8m}$ levels the amount of bins with larger values of $\Delta\bar{\theta}_v$ increases, as a consequence
of being further up from the reference level.

## 5   Turbulence regimes analysis: whole BLLAST dataset

Herein we aim to explore how well the HOST behaves when we use all the night-time dataset from the BLLAST field campaign without any restriction. To achieve this objective, we did the same type of analysis as before. Therefore, the first step is to analyse the differences between all the night-time data wind roses (Fig. 4) and those corresponding to nIOPs (Fig. 3). As
we have seen in Sect. 3 the most remarkable feature is the high frequency of the western flows when the whole dataset is considered. Taking into account that many storms and frontal systems were from this direction (Barneoud et al., 2010), the analysis is done splitting the data between two sets of wind directions: the first group includes winds coming from directions





ranging from $235°$ to $45°$, referred as NW's, including southwest to west, northwest, north and northeast directions; the second group includes the remaining directions, from $45°$ to $235°$ referred as SE's, including northeast to east, southeast, south and southwest directions, which are mostly due to the local topographic effects (Sect. 3).

### 5.1 Turbulence relationships

As in Sect. 4.1, we analyse the turbulence intensity relationships, but herein we focus on the whole night-time dataset without differentiating the fair weather nights to the non-fair weather nights. The analysis is done splitting the data into two main directions, flows coming from the NW's and the SE's directions (Fig. 8). For the NW's winds the relationship between the $V_{TKE}$ with $V$ is linear, so the HOST pattern cannot be appreciated (Fig. 8a). Contrary to this, for the SE's directions the $V_{TKE}$ $vs$ $V$ does resemble to a hockey stick and we can distinguish a $V_T$ for each height (Fig. 8b), which is similar to those

found for the nIOPs but not equal, as the dataset herein includes all the night-time data. In fact, most of the SE's situations correspond to nIOPs (between a $63\%$ and a $70\%$ of them), so the hockey shape reproduced in Fig. 8b is very similar to that shown in Fig. 5. Indeed, this plot corresponds mostly to fair weather nights when stably stratified and near-neutral stratifications occurred and both regimes (regimes 1 and 2) could be distinguished. In contrast, situations corresponding to NW's are tied to frontal systems, cloud skies or cyclonic situations, which substantially modify the boundary layer ideal structure. Specifically,

in these situations the surface may not have been warmed much during the day so nocturnal cooling effects were much more limited, then, the low-wind speed region of the HOST pattern is rarely reached and therefore regime 1 is not well illustrated in the $V_{TKE}$ $vs$ $V$ relationship. For the NW's wind directions, the turbulence seems to be mainly driven by bulk shear, and the HOST turbulence intensity relationship may not be valid in those cases.

### 5.2 Box plots analysis

The box plot representation details are the same as previously explained in Sect. 4.2. In this case, as the data do not correspond only to fair weather conditions, the amount of outliers and the whisker length, especially for higher levels, is considerably greater than for the nIOP situations (Fig. 9 and 10 vs Fig. 6). For this reason, we further examine the turbulence relationships separately for lower (Skinflow mast) and higher levels (Valimev tower). Thereafter we apply two extra conditions to distinguish the reason why the turbulence intensity might have been higher or lower than expected. These points are outliers which have

a very high $V_{TKE}$ value, due to a quick change of the horizontal wind speed, $\sim 1 \, \mathrm{m \, s^{-1}}$, between consecutive measurements (green crosses in Figs. 9 and 10) or to a shift in the wind direction of more than $\sim 75°$ between consecutive measurements (red asterisks in Figs. 9 and 10). These points represent enhanced turbulence intensity for a certain period, temporal average of 5 minutes, due to either wind speed shear or directional shear (Mahrt et al., 2013). Black dots are outliers not attributed to any of these shears, typical of regime 3, associated with top-down turbulent events.

In the following subsections we discuss the box plots for the Skinflow mast levels that are represented in Fig. 9 and the box plots for the Valimev tower levels in Fig. 10.



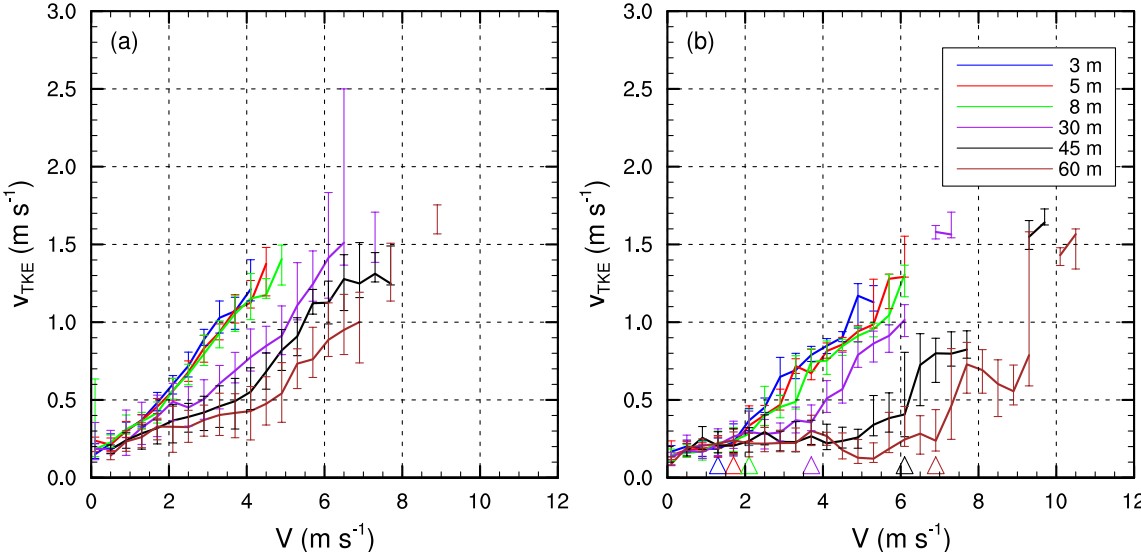

**Figure 8.** Relationships between the bin-averaged turbulence strength $V_{TKE}$ and the wind speed $V$ for the whole BLLAST field campaign for the main considered directions: (a) wind coming from 225° to 45° (included), named NW's, and (b) winds coming from 45° to 225° (included), named SE's. Each line represents the median within each $0.4\ \mathrm{m\ s^{-1}}$ $V$ bin. The vertical lines represent the deviation for each $V$ bin with the low limit corresponding to Q1 and the upper limit to Q3. The threshold wind speed $V_T$ is marked with a triangle with the heights colours in (b).

### 5.2.1 Skinflow mast levels

As seen in subsection 5.1, it is not expected that the relationship between $V_{TKE}$ and $V$ for the NW's directions will behave as the HOST predicts (Fig. 8), therefore, for these directions, a similar behaviour is seen for $z_{3m}$ (Fig. 9a), $z_{5m}$ (Fig. 9b) and $z_{8m}$ (Fig. 9c) levels, where the turbulence strength increases near-linearly with V, and the first regime is not present. The whisker

5    length is small except for lower and higher wind speeds at $z_{8m}$, hence the dispersion is low and most of the points are close to the median, so that the resulting relationship is fairly linear. There are few outliers in $z_{3m}$ and $z_{5m}$ since the surface smooths the quick shifts of wind speed and direction (Mahrt, 2011). The number of outliers increases at $z_{8m}$, many of them caused by wind speed and direction shear (green crosses and red asterisks).

For the thermally driven flows the HOST pattern is clearly identified following both the whiskers and the median values

10    of Fig. 9 d, e, f. There is also a large amount of enhanced turbulence points at all levels triggered mostly by wind speed and directional shear. In addition, for regime 1, there is a large amount of black outliers close to the whiskers that can be related to regime 3, as top-down turbulent events associated to orographic effects are expected to occur from this direction. The IQR range tends to be small except for higher wind speeds, and the whiskers tend to be relatively close to the median. However,



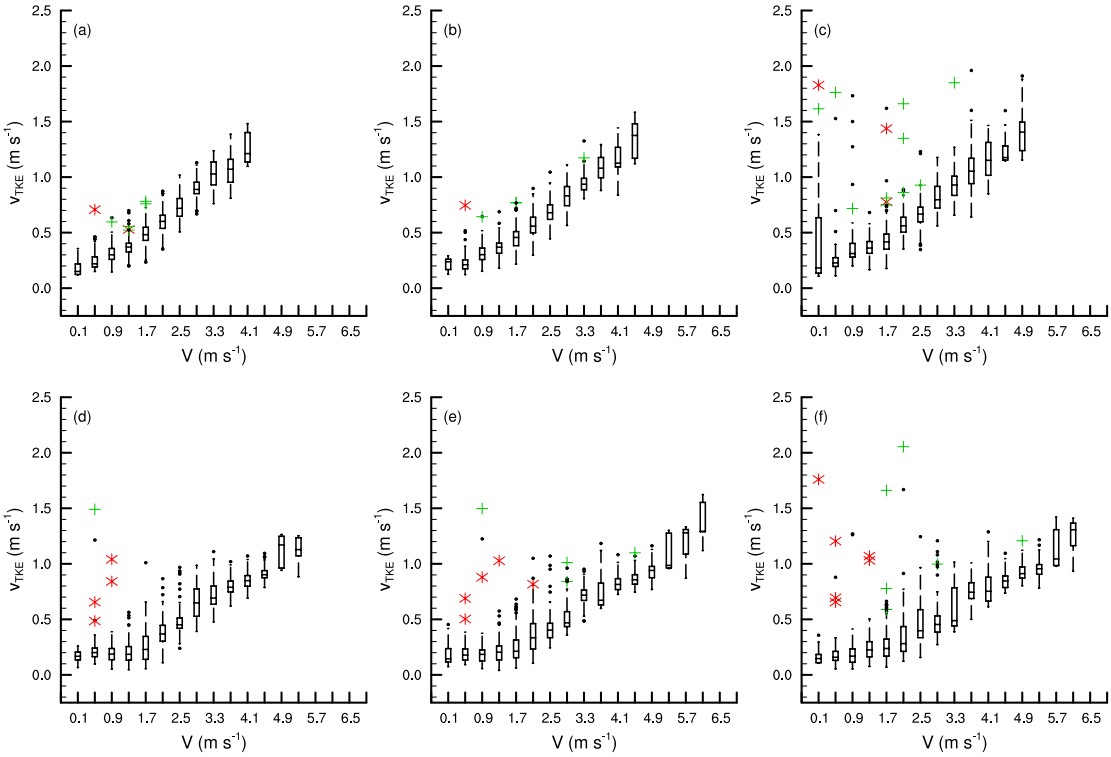

**Figure 9.** Box-plot of the relationship between the bin-averaged turbulence strength $V_{TKE}$ and the wind speed $V$ for the Skinflow night-time dataset for the whole BLLAST field campaign. The top row figures correspond to the NW's direction range for (a) NW's at $z_{3m}$, (b) NW's at $z_{5m}$, and (c) NW's at $z_{8m}$ and the bottom row corresponds to the SE's directions, (d) SE's at $z_{3m}$, (e) SE's at $z_{5m}$ and (f) SE's at $z_{8m}$. The outliers represent a turbulence enhancement due to a quick variation of the wind speed (greater than $1 \, \mathrm{m \, s^{-1}}$) (green crosses), due to a sharp change in the wind direction (greater than $75°$) (red asterisks) or due to any other factor (black dots).

for wind speeds close to $V_T$ the boxes and whiskers are longer, as the corresponding bin-averaged velocity includes turbulence intensities corresponding to the two different turbulence regimes, 1 and 2, with the later having greater $V_{TKE}$ than the former.

This large amount of outliers present in the SE's directions (Fig. 9 d, e, f) is a remarkable difference from what was observed in Sun et al. (2012) and in nIOPs box plot analysis (Fig. 6), where little dispersion was found for each $V$ bin. This increase in 5 the dispersion of the turbulence intensity values for a fixed $V$ may be associated to the topography influence and to the weather conditions, revealing an increase in the amount of non-stationary flows. While CASES-99 was located in a nearly flat terrain, BLLAST is a highly topographically influenced area, where, in addition, the presence of storms and low pressure systems affecting the region is common.



### 5.2.2 Valimev tower levels

For the Valimev tower we have made the same study as before, taking also into account the same main wind directions, NW's and SE's. One of the main results is related to both the IQR range and the whiskers length, for NW's and SE's, they both increase with the height of the measurement, indicating a wide variety of turbulence intensity cases (Fig. 10).

Similarly as in the Skinflow levels, the HOST pattern for the NW's directions cannot be appreciated, whilst for the SE's the turbulence intensity does follow the HOST pattern. For regime 1 in SE's (Fig. 10 d, e, f), even though the boxes are small, there is a great number of outliers, many of them very close to the whiskers which would be corresponding to regime 3. Turbulence enhancement due to wind speed shear dominates over wind directional shear at these levels. Many other outliers (black dots) close to the whiskers in regime 1 are identified as top-down turbulent events (e.g. Kelvin- Helmholtz instabilities, gravity waves

or other intermittent turbulence events).

Overall, for high wind speeds the bin average method does not give good results, as the number of data is not significant enough, hence some absent bins, despite this the trend of regime 2 can be observed. The bins above and below $V_T$ also show greater boxes and whiskers, indicating the inherent uncertainty in determining the real $V_T$ value.

### 5.3 Turbulence regimes and thermal stratification

To further understand the HOST pattern, as previously done in Sect. 4.3 for the nIOPs, we explore the virtual potential temperature gradients dependence of the turbulence relationship for the whole BLLAST dataset. We also separate the data depending on the two wind directions considered, NW's (Fig. 11a) and SE's (Fig. 11b). We only use data from the Skinflow mast, and because of the results among the different levels, $z_{3m}$, $z_{5m}$, $z_{8m}$ is very similar, we only do the analysis and the discussion for the $z_{5m}$ level.

For the NW's flows (Fig. 11a), turbulent mixing leads to a neutral stratification (red and green lines) and also to unstable stratification (turquoise lines), indicating thermal instability in the flow. Stable stratification (purple line) only appears associated to very low velocities and with relatively large $V_{TKE}$ values. Therefore, the linear turbulence relationship is independent of the temperature gradients. Turbulence is driven by bulk shear, therefore the flow incoming from this direction is such that even for lower wind speeds it carries turbulence, i.e. large eddies, that produce similar $V_{TKE}$ independently of the temperature

gradient and of the stratification of the nocturnal BL.

For the thermally driven flows (Fig. 11b) the behaviour of the turbulence intensity versus the wind speed depending on $\Delta \bar{\theta}_v$ is analogous to that found for the nIOPs (Sect. 4.3). Regime 1 is mostly associated with strong positive temperature gradients (purple lines) while near-neutral (green line) and unstable (red line) stratification is characteristic of regime 2 and 3. Indeed, the characteristic events of regime 3, can overturn the typical nocturnal positive temperature gradients.





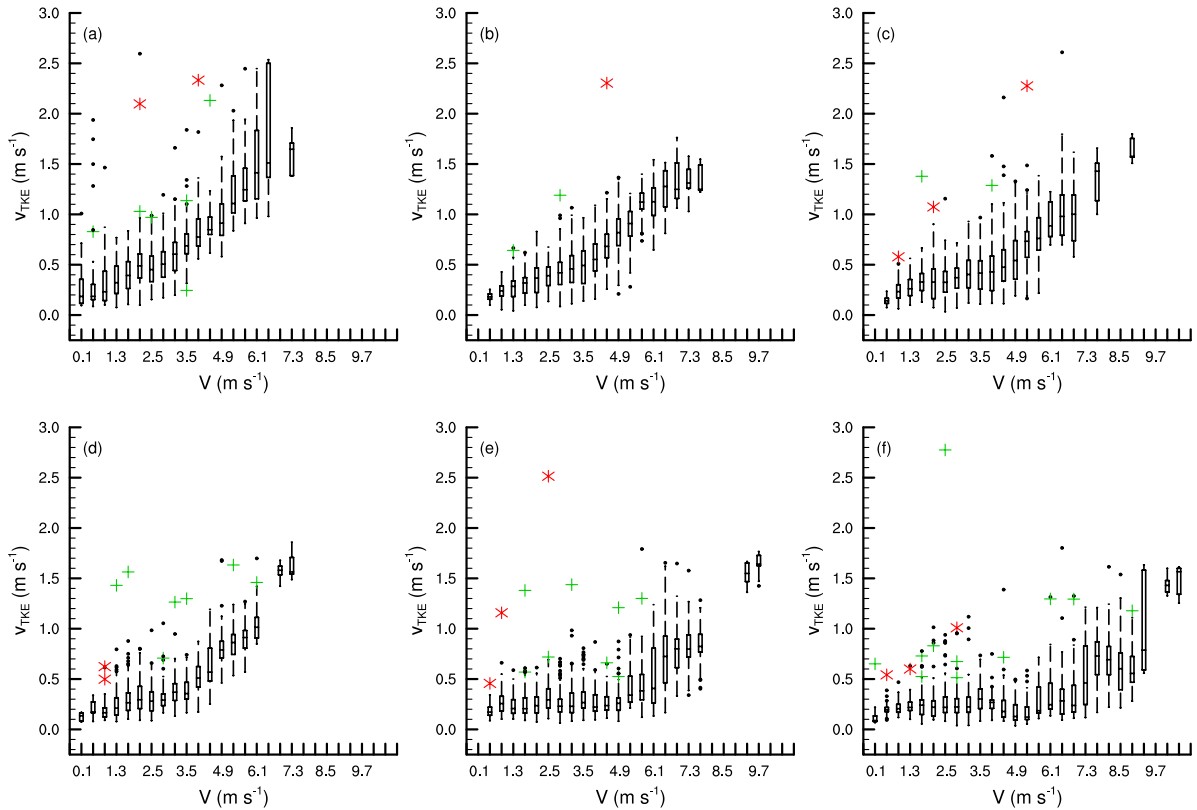

**Figure 10.** Box plots of the relationship between the bin-averaged turbulence strength $V_{TKE}$ and the wind speed $V$ for the Valimev dataset for the whole BLLAST field campaign. Top row figures correspond to the NW's direction range for (a) NW's at $z_{30m}$, (b) NW's at $z_{45m}$, and (c) NW's at $z_{60m}$ and the bottom row corresponds to the SE's directions, (d) SE's at $z_{30m}$, (e) SE's at $z_{45m}$ and (f) SE's at $z_{60m}$. The outliers represent a turbulence enhancement due to a quick variation of the wind speed ($\Delta V > 1 \text{ m s}^{-1}$) (green crosses), due to a sharp change in the wind direction ($\Delta dir > 75°$) (red asterisks) or due to any other factor (black dots).

## 6  Transition between regimes and turbulence intermittency

Associated with the transition between the three turbulence regimes, Sun et al. (2012) defined three turbulent intermittency categories: A, B and C (Fig. 1). Category A corresponds to the transition between regime 1 and regime 2, and vice versa, the wind speed oscillates between both regimes generating intermittent turbulence. Category B corresponds to the enhancement of turbulence within the regime 1 caused by atmospheric disturbances that increases local turbulence and may reduce the local stability, even inducing some intermittency. Category C refers to top-down turbulent events or upside-down boundary layers that enhance downward turbulence into a stable environment, generating regime 3. Since the HOST theory and the related intermittency categories were defined for the SBL, here we only consider the nIOPs and we illustrate the previous categories selecting appropriate examples.





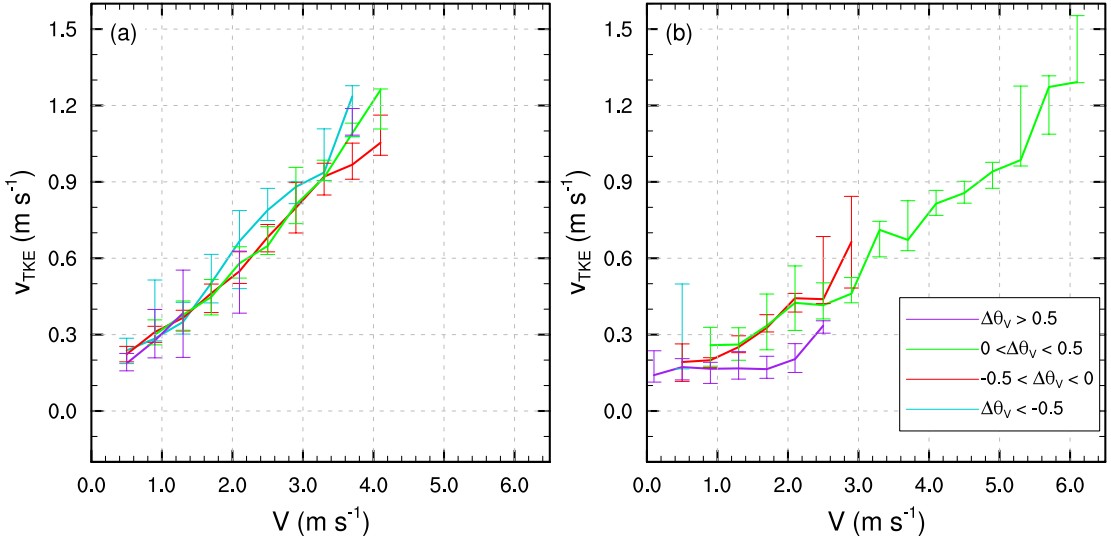

**Figure 11.** Relationship between the bin-averaged turbulence strength $V_{TKE}$ and the wind speed $V$ during the BLLAST campaign nIOPs as a function of potential temperature difference intervals, defined as, $\Delta \bar{\theta}_v = \bar{\theta}_v(z) - \bar{\theta}_{v_{z_0}}$, where $\bar{\theta}_{v_{z_0}}$ is the virtual potential temperature of reference at 2 m, and $\bar{\theta}_v(z)$ the virtual potential temperature at the height of $z_{5m}$ for: (a) NW's direction range and (b) SE's direction range. The wind speed threshold $V_T$ for $z_{5m}$ is marked with a black triangle in (b).

## 6.1 Turbulence intermittency: Categories A & B

During the BLLAST field campaign the occurrence of katabatic winds coming from the south valleys was quite common (Román-Cascón et al., 2018). They are usually generated after the near calm situation of the evening transition, first as shallow drainage flows, which later are broken by the arrival of a larger and deeper mountain-plain flow with greater values of V.

The transition between them can cause wind speed oscillations between regime 1 and 2, and shallow wind oscillations within regime 1 leading to turbulence intermittency of category A and B respectively. During the BLLAST field campaign both type of transitions are detected, either by the Valimev tower or by both the Valimev and the Skinflow levels. In almost every nIOP one of the different ways of achieving this category of turbulence intermittency is observed. For this study we select the period between 1900 UTC and 2200 UTC during on the night of the 2nd of July of 2011 (nIOP09) (Román-Cascón et al., 2015), as

it represents better the intermittency of both of these categories, as shown in Fig. 12. The categories can also be found during other nIOPS.

The 2nd of July of 2011 (IOP10) was characterised by an anticyclonic situation over the south of France, leading to weak surface gradients and to near calm conditions close to the surface at the evening time. Thus, at 1900 UTC the wind speed values are around $0.5 \, \mathrm{m \, s^{-1}}$ (Fig. 12a). This situation is, according to Román-Cascón et al. (2015), the foremost scenario for

"the appearance of SDFs with a marked SSE-SE component, where stronger winds were encountered at lower levels with



maxima close to the surface (around 2-3 m above ground) and the wind intensity decreasing with height above the maximum". This is the picture of a slight SDF flowing through the nearby gully (Román-Cascón et al., 2015; Nauta, 2013) measured by the Skinflow mast and the Valimev towers, although in Fig. 12 only the $z_{30m}$ level of Valimev tower is shown, for better clarity. The arrival of this SDF promotes the establishment of a surface temperature inversion (Fig. 12b), with a larger decrease in

temperature at the lowest levels. The SDF stage ends around 2025UTC with the arrival of a stronger and deeper wind (Fig. 12a) from the southeast, and the MPF stage starts. This increase in wind is more noticeable at high levels, leading to the breaking of the SDF and the potential virtual potential temperature homogenisation (Fig.12b).

As Román-Cascón et al. (2015) showed, two different gravity wave events occurred during the SDF stage and the arrival of the MPF. Both events are examples of turbulence intermittency categories. The first one corresponds to a wave of almost four

cycles with a 20-25 min period between 19:00 and 2025 UTC approximately. This is an example of category B of turbulence intermittency, as the oscillations in wind speed occur for wind speeds below the threshold of each level (see the $V_T$ for each height marked with a coloured horizontal dashed line in Fig. 12a), within regime 1. The turbulence can be seen clearly for $z_{2m}$, $z_{3m}$, $z_{5m}$ and $z_{30m}$ (see arrow cat. B in Fig. 12a). Román-Cascón et al. (2015) states that the second event "is characterised by several wind speed oscillations of shorter periods with two notable cycles of greater amplitude from 2030 to 2130 UTC". The

wind speed oscillates around the threshold, as an example of category A. There is a transition from regime 1 to 2 and viceversa at $z_{3m}$, $z_{5m}$, $z_{8m}$ and $z_{30m}$ (see arrow cat. A in Fig. 12a). In addition, when the wind speed is above the threshold, the virtual potential temperature vertical gradients are reduced (Fig. 12b), as it is expected in regime 2.

## 6.2  Turbulence intermittency: Category C

Category C turbulence intermittency refers to the transition between regimes 1 and 3 in the HOST model (Fig. 1). It is originated

in very stable boundary layers when the main source of turbulence is elevated, and is temporally detached from the surface but intermittently coupled to the surface generating bursts of downward turbulence (Blumen et al., 2001). The origin of this elevated turbulence can be related to the presence of Kelvin-Helmholtz instabilities, density currents, gravity waves, low level jets or any other meteorological phenomena that can induce turbulence intermittency.

Usually, Cat. C turbulence intermittency is characterised by a decrease of V, an increase in the variability of $w$ with height,

and negative values of the skewness, $S$, (Sun et al., 2012; Blumen et al., 2001), meaning a downward transport of turbulence (Mahrt and Vickers, 2002).

In order to illustrate an example of Cat. C turbulence intermittency we select an event taking place between 0300 and 0400 UTC of the 26 of June of 2011, nIOP05. From Fig. 13a we can observe a progressive wind speed decrease in the higher levels of the Valimev tower between 0310 and 0350 UTC (see black rectangle Fig. 13a), associated with an increase on the

vertical wind speed variability at 60 m, which has the maximum between 0345 and 0355 UTC (Fig. 13b). This maximum is intensified with height and its time delay as the levels are closer to the surface is due to its downward propagation nature. This downward turbulence transport is also seen with the negative $S$ values, show in Fig. 14 for $z_{8m}$ and $z_{60m}$. The period where the skewness is more negative agrees well with the period at which the larger oscillations of $w$ take place (0350 UTC). All these characteristics could indicate a category C type of turbulence intermittency event, representing occasional mixing found within



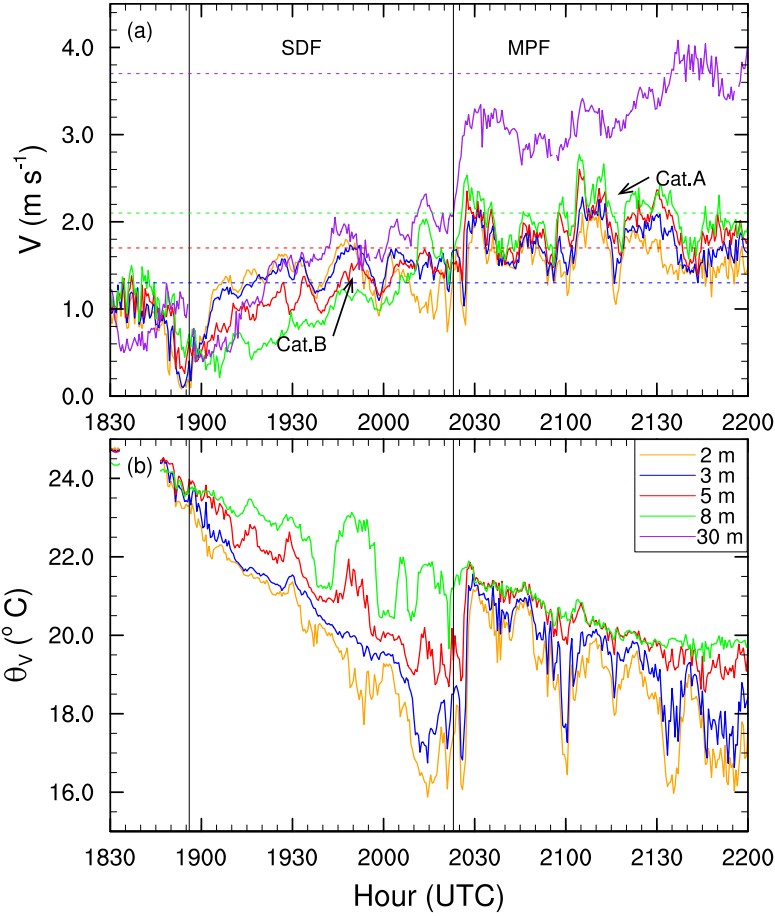

**Figure 12.** Time series of (a) horizontal wind speed and (b) virtual potential temperature for the 2 of July (nIOP09) during the BLLAST field experiment for $z_{2m}$, $z_{3m}$, $z_{5m}$, $z_{8m}$ and $z_{30m}$. The shallow drainage flow (SDF) stage is indicated between 1855 and 2020 UTC, and a consecutive mountain-plain wave (MPF) stage thereafter. Both stages originated transitions between regime 1 to regime 2 (category A), and enhancement of turbulence within regime 1 (category B). The potential temperature is not represented for $z_{30m}$ because there is only thermocouple measurements for the Skinflow mast ($z_{2m}$-$z_{8m}$). The wind speed threshold for several levels are marked with the dashed line to better illustrate the transitions, whilst some transitions between regime 1 and regime 2 are marked with an arrow. The data shown herein is the 30 s temporal average from sonic anemometers with a frequency of 10 Hz.

the weak turbulence and a downward transport of turbulence (Mahrt and Vickers, 2002), both signals of a possible top-down turbulence event. It is relevant to note that this event is not strong enough to arrive to the lower levels of the Skinflow mast.

Cat. C turbulence intermittency does not occur as often as Cat. A and Cat. B, as the number of points within regime 3 in Figs. 6, 9, and 10 shows. These points could sometimes correspond to a top-down event, but not always, since the regime 3 can be achieved through accelerations and sharp wind direction shifts (Mahrt et al., 2013).



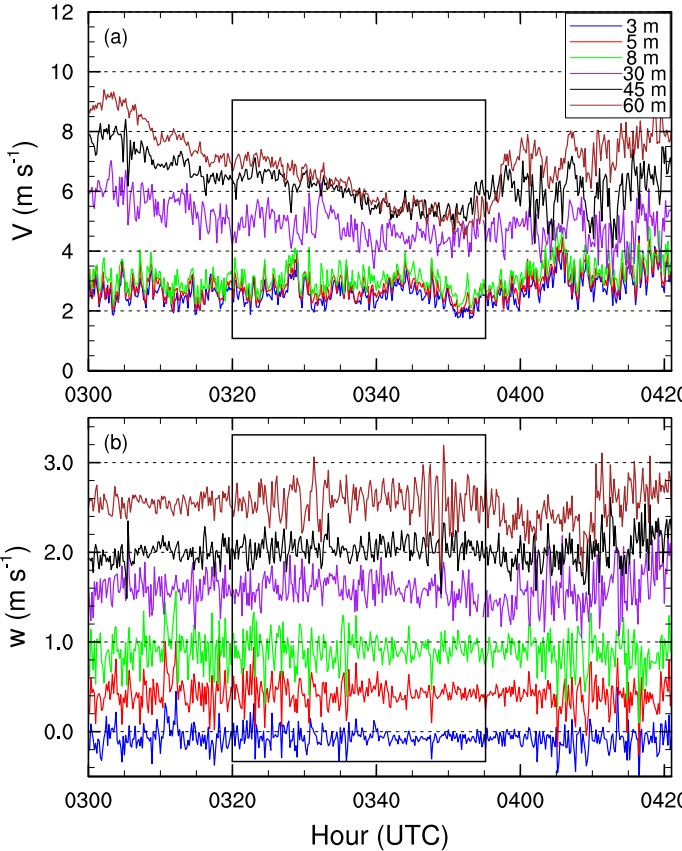

**Figure 13.** Times series of (a) horizontal wind speed $V$ and (b) vertical wind speed, $w$ (incremented $0.5\ \mathrm{m\ s^{-1}}$ for an easy comparison) for 26 June 2011 at $z_{3m}$, $z_{5m}$, $z_{8m}$, $z_{30m}$, $z_{45m}$ and $z_{60m}$. The black rectangle localises the top-down event. The data shown herein is the 10 s temporal average from sonic anemometers with a frequency of 10 Hz.

# 7 Conclusions

In this study we have analysed the HOST theory proposed by Sun et al. (2012) using the nocturnal data set of the BLLAST campaign, that took place at the "Plateau de Lannemezan", an area located north of the central French Pyrenees. The main objective has been to explore the influence of the orography and the weather conditions during the BLLAST campaign in the HOST pattern that was originally defined for a relatively flat area and for fair weather conditions, the predominant conditions of the CASES-99 field campaign. Therefore, the analysis has been done separating nights into two types: nocturnal IOPs (nIOPs) and the whole nocturnal dataset.

For the nIOPs of the BLLAST campaign the HOST theory is found to be valid and the turbulence relationships show the different turbulence regimes with a wind speed threshold for each height which locates the transition between the weak wind





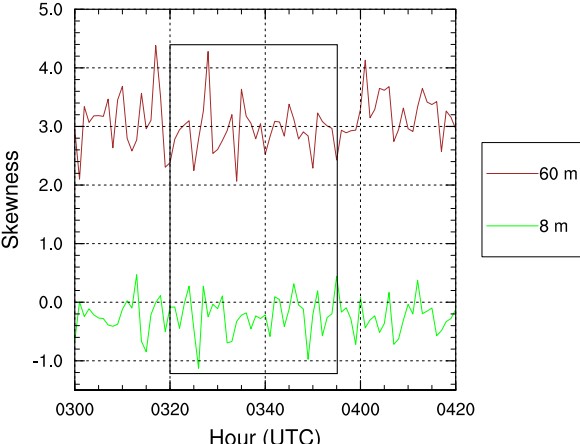

**Figure 14.** Time evolution of skewness of vertical wind component measured between 0300 and 0400 UTC on 26 of June during the BLLAST field experiment for the levels $z_{3m}$ and $z_{60m}$. For the sake of clarity, the 60 m level skewness measurements are increased by $3 \mathrm{~m~s}^{-1}$. The black rectangle localises the top-down event. The data shown herein is the $1 \mathrm{~min}$ temporal average from sonic anemometers with a frequency of 10 Hz.

and weak turbulence intensity generated by local shear, regime 1, and the strong turbulence intensity generated by bulk shear, regime 2. This wind speed thresholds are lower than in Sun et al. (2012) so the bulk shear needed to achieve regime 2 is lower as well, due to the larger roughness length of the mountainous terrain linked with the incoming southern flow. Similar results were found by Mahrt et al. (2013) and Bonin et al. (2015) for other locations. These nights are characterised by clear skies and

intense radiative cooling where the predominant flow comes from the southeast quadrant corresponding to the arrival of shallow drainage flows to the lower levels and mountain plain flows later on. These flow circulations generated by the orography lead to turbulence enhancement and transitions between regimes, modifying the ideal HOST pattern.

Instead, when the turbulence relationship is analysed for the whole night dataset, results are significantly different, and they become wind direction dependent. The HOST pattern is observed for thermally driven flows, from wind directions between

$45°$ and $235°$ (SE's) but it cannot be appreciated for winds coming for the other directions (NW's). For the NW's flows, the turbulence relationship is almost linear and independent of the vertical temperature gradients. Indeed, when considering the whole dataset the frequency of winds coming from westerly and northerly directions increases representing the mesoscale and synoptic scale meteorological situations, in which the HOST pattern is erased mainly because the stable boundary layer cannot be developed.

To further investigate the turbulence relationship, we used box plots to provide additional information about the turbulence relationships. Comparing the nIOPs and the SE's flows from the whole dataset there are interesting differences: (i) the turbulence intensity tends to increase when we use the whole nocturnal dataset; (ii) the number of outliers increases for the whole nocturnal dataset, and therefore the number of turbulent events associated to transitions between regimes 1, 2 and 3. Several of




the points falling in regime 3 are actually associated with sudden wind speed and wind directional shear transitions, reflecting the non-stationary nature of regime 3.

In addition, when HOST is achieved, near neutral stratification is tied to regime 2, where turbulence is driven by bulk shear. Instead, when $V < V_T$ stable stratification dominates, although negative potential temperature differences are present when
intermittent turbulence events occur.

Finally, the different transitions between the defined HOST regimes, categories A, B and C, are illustrated for some nIOPs using appropriate examples (sect. 6). Results show that the presence of gravity waves associated to shallow drainage flows and mountain plain flows create intermittent turbulence that can lead to a transition between regime 1 and regime 2, an example of category A. When the wind speed is below the threshold velocity. i.e. a category B transition, local shear can be generated by
internal gravity waves. These oscillations are typically originated within the shallow drainage flow (Fig. 12). Category C may also appear when turbulence created by wind speed or directional shear is diffused downward, toward a stable environment, as shown in Fig. 13 and Fig. 14.

Present and future work related to the knowledge of the flow circulation in BLLAST site would help to the full understanding of the obtained results in this study. In addition, a deeper and longer-term study of the HOST theory in other locations,
considering mountain influenced sites and complex terrain areas would be desirable so that our outcomes could be generalised.

*Author contributions.* Marie Lothon supplied the access to all the data set, as well as first hand insight to the BLLAST site and its phenomena. Erik Nilsson did some data treatment for the Skinflow mast and useful and precise advice on conceptual matters. Jielun Sun provided very helpful information on the HOST theory and ideas for improvement on the initial idea. Joan Bech, altogether with everyone provided help on the manuscript structure and general advice on turbulence. Maria Rosa Soler and Mireia Udina played a crucial part in the processes of
crafting this manuscript from scratch with Jesús Yus-Díez, developing the data process, the analysis of the results, and summarizing and expressing it in this article.

*Competing interests.* No competing interests are present

*Acknowledgements.* The work was supported by the Spanish Government through projects CGL2015-65627-C3-2-R and MINECO CGL2016-81828-REDT. We would also to acknowledge for the BLLAST data policy.The BLLAST field experiment was made possible thanks to the
contribution of several institutions and supports : INSU-CNRS (Institut National des Sciences de l'Univers, Centre national de la Recherche Scientifique, LEFE-IDAO program), Météo-France, Observatoire Midi-Pyrénées (University of Toulouse), EUFAR (EUropean Facility for Airborne Research) and COST ES0802 (European Cooperation in the field of Scientific and Technical). The field experiment would not have occurred without the contribution of all participating European and American research groups, which all have contributed in a significant amount (see supports). BLLAST field experiment was hosted by the instrumented site of Centre de Recherches Atmosphériques,
Lannemezan, France (Observatoire Midi-Pyrénées, Laboratoire d'Aérologie). Its 60 m (["Valimev"]) tower was partly supported by the



POCTEFA/FLUXPYR European program. BLLAST data are managed by SEDOO, from the Observatoire Midi-Pyrénées. As well as for the respoinsibles of the instruments and data that we have used: Fabienne Lohou, PI of surface measurements in BLLAST, Frédérique Saïd, Solène Derrien for the 60 m instrumentation and data, Eric Pardyjak and Daniel Alexander for the 8 m ("Skinflow") tower. Also to the P2OA instrumented site, since the Centre de Recherches Atmosphériques of Lannemezan is part of an instrumented platform called Pyre-

5  nean Platform of the Observation of the Atmosphere (http://p2oa.aero.obs-mip.fr). P2OA facilities and staff are funded and supported by the Observatoire Midi-Midi-Pyrénées (University of Toulouse, France) and CNRS (Centre National de la Recherche Scientifique).





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
