# Peer review of "Nocturnal boundary layer turbulence regimes analysis during the BLLAST campaign"

_Atmospheric Chemistry and Physics, 2018_

## Referee Comment (RC1) · Anonymous Referee #1 · 30 Mar 2019

The paper deals with observations recorded during the Boundary-Layer Late Afternoon and Sunset Turbulence (BLLAST) field campaign which took place in 2011 a few kilometres north to the Pyrenean foothill, around the *Centre de Reserches Atmosphériques* in the Lannemezan Plateau. This database has been extensively analysed during the last years with many interested papers published in high impact journals. The present paper shows an original work looking for the application of the HOST (HOckey Stick Transition) theory (Sun et al., 2012) to the BLLAST data, where the presence of heterogeneous terrain and orographic features could modulate the theory. The subject is well introduced and the paper is generally well written and structured, but I think that the next general and specific comments should be taken into account before the manuscript could be accepted. My recommendation is 'Major Revisions'.

**General comments:**

- A point that is not developed either discussed in the paper is the importance of the height of the Low Level Jet (LLJ) for the different events analysed in the work. Depending if you are analysing levels above or below the LLJ the behaviour of the turbulence transport can be different showing if turbulence is connected or not with surface, or if MOST can be used (see for example Grachev et al., 2016). This issue could be connected to the different regimes that are found using the HOST theory and it can be interesting to explore it.

- When it is said that more than 60% of the flows at nIOPs come from SE quadrant and correspond to shallow drainage flows (SDF), did you test that they are really SDF? How shallow? What is the height of the LLJ found? I think that you should analyse this issue in a deeper way.

- Both in the abstract and along the paper you associate the flow coming from NW's to mesoscale or synoptic scales. Y agree with synoptic, but not with mesoscale, or at least not will all the mesoscale; for example, thermally-driven flows producing mountain breezes have their origin in the SE's and they are mesoscale flows. So, this should be revised along the paper.

- You use the data from night-time (sunset to sunrise). As stably-stratified conditions are reached before the sunset, have you done any sensitivity test to what differences can be obtained in the results if you consider for example instead the sunset, the time when sensible heat flux changes sign and becomes negative?

- In section 2 (at the end of page 4) you mention that 5 min. is used to evaluate the turbulent quantities, and you cite some references. I think that it could be

interesting to discuss a bit more the importance of using 5 min. instead of other temporal average (larger or shorter) in your study.

- I find difficult to follow the information given in Figs. 3-4, those where you show the wind roses. This is not the traditional way in which wind roses are represented (see for example Hullin et al., 2019; Fig. 2 for a better representation). By the way, I think this paper can be interesting for your present work and could be referenced. With regards to the information shown in this figures (3-4), I would like the authors to discuss more the differences found in wind direction distribution between Valimev and Skinflow towers, both for nIOPs and night-time whole dataset. For example, SE is clearly predominant for the Skinflow tower heights vs. Valimev for both datasets.

- I think it could be interesting to discuss how do you estimate the intermittency of the turbulence. I think it is not enough explained along the manuscript. Moreover, in the literature there are different definitions of turbulence intermittency, so it is important to know what you are using in the present study.

- References:

- Grachev, A. A., Leo, L. S., Di Sabatino, S., Fernando, H. J. S., Pardyjak, E. R., and Fairall, C. W.: Structure of Turbulence in Katabatic Flows Below and Above the Wind-Speed Maximum, Bound.-Lay. Meteorol., 159, 469–494, 2016.
- Hulin, M., F. Gheusi, M. Lothon, V. Pont, F. Lohou, M. Ramonet, M. Delmotte, S. Derrien, G. Athier, Y. Meyerfeld, Y. Bezombes, P. Augustin, and F. Ravetta: Observations of Thermally Driven Circulations in the Pyrenees: Comparison of Detection Methods and Impact on Atmospheric Composition Measured at a Mountaintop. J. Appl. Meteor. Climatol., 58, 717–740, https://doi.org/10.1175/JAMC-D-17-0268.1 , 2019.

**Specific comments:**

1) Revise the order of the references when you are citing more than one. Generally chronological order should be used and this is not always done in the manuscript (see for example in page 4, lines 13-14; pag. 7, line 9; pag. 14, line 5; …………….).
2) Pag. 2, lines 19-21: some reference could be given in relation with the TTE concept (Zilitinkevich et al., 2007, for example that you already have at the reference list).
3) Pag. 3, line 29: replace pikes by peaks.
4) Pag. 3, line 30: I think it is less than 45 km.
5) Pag. 4, lines 10-11: I do not understand this sentence. Could you please revise it? I do not find any relationship with the phrase that comes next.

6) From my point of view, the information given in pag. 5 (lines 8-16) is difficult to understand as it is, and I think that it is not necessary and could be discarded. Maybe you can reference the papers by Said et al., but not giving the detailed information that comes next. However, I missed some post-processing information of the sonic data. For example, the kind of rotation applied (double rotation, planar-fit?).

7) Pag. 6, lines 16-17. The reference Román-Cascón et al. (2018) is Román-Cascón et al. (2019) and the complete reference at the Reference list is also wrong (for example the title or Journal); below you have the correct one.

8) Pag. 6, lines 28-29: when you say at the lower levels, indicate exactly the levels considered. And in line30, the same for the higher levels.

9) Pag. 7, lines 1-9. In this context, it can be interesting reference the results found by Jiménez et al. (2019).

10) Pag.7, lines 5-7: you relate the occurrences of NW at higher levels with a SBL height below the Valimev tower and strong synoptic forcing. Have you check this point? Have you estimate the SBL height? From my point of view, when strong synoptic forcing is present then the nocturnal ABL height should be weakly stably-strafified and the ABL height should be quite larger than 60m.

11) Pag.7, lines 7-9: I cannot see in Fig. 2a the very small valley you mention al the south of the Skinflow tower. Could you give more information on this gully (slope and its orientation)? It can be quite interesting to know it.

12) Pag. 7, lines 10-13: When you reference the SDF described by Román-Cascón (2015), it is said that it ranges from noon $1^{st}$ July to morning $2^{nd}$ July, including nIOP8. This is wrong. The period analysed in Román-Cascón et al. (2015) ranges approx. from 18:00 to 22:00 UTC on the $2^{nd}$ July (IOP10), and the SDF lasts from 19:00 to 20:30 UTC approx.

13) Pag. 7, line 13: Change $1^{st}$ July by $2^{nd}$ July.

14) Pag. 9, lines 1-2: It is said that the MPF is from SW. However, in pag. 6, line 31 it is mentioned that MPF comes from the SE quadrant. Could you explain this contradiction?

15) Pag. 10, lines 30-32 and pag. 11, lines 1-2: A comment in the line of that done in comment 10; you justify that regime 2 does not behave as HOST for the 60m for the largest winds because this height could be above the top of the SBL. However, I would not expect this just for the highest winds, where the NBL height can be higher due to mechanical turbulence generated for stronger winds.

16) Pag. 12, lines 13-15: "the turbulence intensity can be enhanced due to the presence of coherent structures." My question is: for what range of wind speed do you think it is more relevant the presence of coherent structures (CS) and why? In relation with this question, in this same page, lines 18-21, it would seem that you have more presence of coherent structures (internal gravity waves for example??) for larger winds, so for near-neutral conditions. Do you really think that CS are related to NBL more than SBL? Could you please clarify this in the discussion?

17) Pag. 12, line 26: 'big difference'. Could you explain these differences? In line 28, when you use 'In addition', it seems that you are going to discuss about Skinflow tower, but you are referring to Valimev tower. Clarify it, please.

18) Pag. 16, lines 6-7: 'There are few outliers in z3m and z5m since the surface smooths the quick shifts of wind speed and direction'; could you please explain better this sentence?

19) Pag. 17, lines 3-8: In this paragraph you are discussing the presence of outliers in the SE's directions, and in part it is related to the presence of storms and low pressure systems affecting that region. I think that at least low pressure systems are related to NW's not to SE's directions.

20) Pag. 19, line5: when you mention 'atmospheric disturbances', at what scales are you referring to? Are internal gravity waves or other submeso motions important in this context?

21) Pag. 19, lines7-9: why don't you consider SW's instead of nIOPs to illustrate the intermittency categories? In fact, you can have suitable conditions (SBL) even when an IOP is not defined in BLLAST.

22) Pag. 20, lines 1-11. The paper from Roman-Cascón et al. (2019) does not use BLLAST data (this is done in Román-Cascón et al. 2015), although they characterize the thermally-driven flows at the BLLAST site. So please cite properly both papers.

23) Pag. 20, lines10-11: 'The categories can also be found during other nIOPS.'. Please indicate explicitly those nIOPs.

24) Pag. 21, lines19-23: Can you indicate any references at the end of this paragraph?

25) Pag. 21, Fig. 12b: I cannot find the purple line, corresponding to the 30m height.

26) Pag. 21, Figure 12 caption: I understand that category A is related to MP flow and category B to SDF, so it would be clearer if you state: Both stages….enhancement of turbulence …and transitions between reg 1 to reg 2 respectively.

27) Pag. 24, line 12-13: 'mesoscale and synoptic scale meteorological situations'. According to this statement, mountain breezes, SE's are not a mesoscale flow?

28) Pag. 24, lines 17: 'whole nocturnal dataset' or it should say 'whole SE's nocturnal dataset'?

29) Pag. 25, line 6: after C you could add 'related to turbulence intermittency'.

30) Pag. 25, lines 9-10: Could you explain how local shear can be generated by internal gravity waves?

31) Pag. 27, lines 33-35: This paper is already published and the right journal is 'Atmospheric Research' and not 'Atmospheric Environment'. Please change it.

- References:

- Jiménez, M. A., Cuxart, J., and Martínez-Villagrasa, D.: Influence of a valley exit jet on the nocturnal atmospheric boundary layer at the foothills of the Pyrenees, Q. J. Roy. Meteor. Soc., 145, 356–375, 2019.

- Román-Cascón, C., Yagüe, C., Arrillaga, J., Lothon, M., Pardyjak, E., Lohou, F., Inclán, R., Sastre, M., Maqueda, G., Derrien, S., Meyerfeld, Y., Hang, C., Campargue-Rodríguez, P., and Turki, I.: Comparing mountain breezes and their impacts on CO2 mixing ratios at three contrasting areas, Atmos. Res., 221, 111–126, 2019.

---

## Referee Comment (RC2) · Anonymous Referee #2 · 3 May 2019

The manuscript contains good science. The reader will learn important aspects of the hockey stick relationships between turbulence and mean winds. The most relevant ones are the dependence on the upwind terrain characteristics and the role played by the thermal gradients. I also found the characterization of the different types of intermittent events quite interesting, although it is a bit too simplistic. My main concerns with the paper regard its presentation. In that sense, I have two major suggestions and a few minor ones. I am classifying it as a major review because it may affect the manuscript largely.

MAJOR SUGGESTIONS:

1. I do not think the entire section showing how the IOP cases behave is necessary. Very similar plots are shown in the following section, specially given that one of the wind

directions considered coincides largely with the occurrence of IOP cases. It seems to me that no major conclusions are drawn from the IOP analysis alone. In fact, when reading that initial section of the paper, I was a bit disapointed, because I felt I was just reading an assessment on how the findings of Sun et al. (2012) apply to a different dataset. In the following section, on the other hand, there is novelty, related to the different hockey sticks and their dependence on the upwind terrain characteristics. Why not start with that, then? Moreover, why to keep the analysis on the IOP cases?

2. I would like to see some deeper development on the intermittency analysis. Maybe some statistics on how common each classes are, if they are favores under some type of external condition (large-scale wind direction) and how they affect the thermal structure.

MINOR SUGGESTIONS

3. Page 5, lines 10-12. It is not necessary to tell the rotations you applied to the different sonic data. These are associated to the sonic deployment, which is specific to your experiment;

4. I would drop Fig. 3, for reasosn detailed in point 1, above.

5. Page 12, lines 7-12. This type of explanations belong at the figure caption;

6. Page 12, line 33. It is very difficult to understand what "too deep to calculate the bulk potential temperature difference" means. Usually it is easier to evaluate gradients over deeper layers, as they are larger.

7. Page 14, lines 8-10. This issue can be easily solved by using the thermal gradient dT/dz instead of temperature difference alone.

Please note that although the comments above refer to the IOP analysis (which I suggested being removed from the paper) they also apply to the analysis of the whole dataset, which shows the same types of plots.

---

## Author Comment (AC1) · 19 Jun 2019

Herein attached is the response to the referee #1 comments. We have appended the manuscript with all the changes on it. The struck through red lines represent the delated old text, whereas the blue text represents the new text that appears in the revised manuscript.

Please also note the supplement to this comment:
https://www.atmos-chem-phys-discuss.net/acp-2018-1300/acp-2018-1300-AC1-supplement.pdf

2019.

**Supplement:**

This document answers the questions posed and the point of view on several aspects of the paper herein discussed. We would like to acknowledge the valuable comments by the referee.

**General comments.**

**1) A point that is not developed either discussed in the paper is the importance of the height of the Low Level Jet (LLJ) for the different events analysed in the work. Depending if you are analysing levels above or below the LLJ the behaviour of the turbulence transport can be different showing if turbulence is connected or not with surface, or if MOST can be used (see for example Grachev et al., 2016). This issue could be connected to the different regimes that are found using the HOST theory and it can be interesting to explore it.**

Response: Thank you for your comments and suggestions. We try to answer your requirements using the previous work done by different authors. Nilsson et al., (2014) analyzed the wind profile using data from the Skinflow tower during 10 IOP nights, showing that all of them, except one, displayed reverse wind gradient or non-monotonic wind profile with wind direction from about 120 to 180 degrees. This result as Román-Cascón et al., (2015) indicated corresponds to shallow drainage flows SDFs exhibiting a wind maximum around 1.5-2 ms$^{-1}$ close to the surface at 2-3 m a.g.l., and also a larger scale down slope wind coming from the south and channeled through the valleys where Pyrenees mountain range is located. The maximum of this larger scale thermal wind is around 80 m a.g.l., as results of WRF simulations (Román-Cascón et al., 2015). In addition, the organization of the flow in the Aura Valley as Jimenez et al., (2019) indicate generates a valley exit jet close to the midnight, which propagates through the foothills while its characteristics (speed and height) decrease. A maximum wind speed of about 5–10 ms$^{-1}$ from the southern sector is found in Lannemezan between 50 and 200 m above the ground, depending on the features of the mesoscale winds (Jiménez et al., 2019).

All low-level jets considered above can be related to the three turbulent regimes. In this way, as Sun et al., (2012) stated we can consider two possibilities:
a) The maximum wind of a LLJ is less than the threshold value for its height above the ground, but the local shear below the LLJ is considerable, then moderate turbulence can be generated, diffusing downwards, especially when the vertical temperature gradient decreases with height, this situation corresponds to the turbulence regime 3.
b) The maximum wind of a LLJ is greater than the threshold value for its height above the ground. This case corresponds to the turbulence regime 2.
In addition to other factors, also the occurrence of the different regimes can be linked to the presence of LLJs.

This answer is included in several of the paragraphs of the new Sect. 3, between pages 6-9.

**2) When it is said that more than 60% of the flows at nIOPs come from SE quadrant and correspond to shallow drainage flows (SDF), did you test that they are really SDF? How shallow? What is the height of the LLJ found? I think that you should analyse this issue in a deeper way.**

Response: This comment allowed us to identify that there was a mistake so the sentence is re-written. Fig 3 shows that a 25% of the flows at the lower levels (Skinflow tower) come from the southeast quadrant, which correspond to shallow drainage flows, formed after sunset due to local small slopes located in the foothills of the Pyrenees (Román-Cascón et al., 2015). These authors tested that up to 4 days of the BLLAST campaign present SDFs after the near calm period of the afternoon, exhibiting a wind maxima, around 1.5-2 $ms^{-1}$ close to the surface, in the first 5 m a.g.l. At higher levels of the Valimev tower wind roses show a large fraction of winds coming from the southeast quadrant (35 % of the data), which are associated to the larger scale mountain plain circulation. The maximum of this thermal wind is located around 80 m a.g.l. as a result of WRF simulations indicated by Roman-Cascón et al., 2015. In addition, as reported by Jimenez et al. (2019), the organization of the flow in the Aura Valley generates a valley exit jet close to midnight, which propagates through the foothills while its speed and height decrease. A maximum wind speed of about 5-10 $ms^{-1}$ from the southern sector is found in Lannemezan between 50 and 200 m above the ground, depending on the features of the mesoscale winds (Jiménez et al., 2019).

As in the previous answer, this is included in several of the paragraphs of the new Sect. 3, between pages 6-9.

**3) Both in the abstract and along the paper you associate the flow coming from NW's to mesoscale or synoptic scales. I agree with synoptic, but not with mesoscale, or at least not will all the mesoscale; for example, thermally-driven flows producing mountain breezes have their origin in the SE's and they are mesoscale flows. So, this should be revised along the paper.**

Response: According to your comment we have removed the word mesoscale in the sentences related to flows coming from NW's.

**4) You use the data from night-time (sunset to sunrise). As stably-stratified conditions are reached before the sunset, have you done any sensitivity test to what differences can be obtained in the results if you consider for example instead the sunset, the time when sensible heat flux changes sign and becomes negative?**

Response: We have not done any sensibility test - regarding this point we have followed the same approach as Lothon et al., (2014) and Blay-Carreras et al., (2014) about the sunrise and sunset time during this period and at this area. Considering your comment, we have added the following explanation in Sect. 4, page 11, between lines 20-23: "An alternative approach to calculate the stratification conditions could be to use the thermal gradient instead. However, throughout the paper we have considered the bulk potential temperature difference method described in Sun et al., (2016) which also provides good results and allows for a more consistent comparison within the Hockey-Stick theory framework."

**5) In section 2 (at the end of page 4) you mention that 5 min. is used to evaluate the turbulent quantities, and you cite some references. I think that it could be interesting to discuss a bit more the importance of using 5 min. instead of other temporal average (larger or shorter) in your study.**

Response: We have taken into account your comment and introduced it in the text, in Sect. 2, page 5 line 1-7: "[…] reducing the averaging period, or using other methods as multi-resolution decomposition or wavelet, might potentially eliminate the contribution of non-turbulent mesoscale motions to the calculated turbulence quantities (Terradellas et al., 2005; Udina et al., 2013; Ferreres et al., 2013; Soler et al., 2014). However, to obtain a broad picture of the patterns, a constant averaging time of 5 min. has seemed sufficient. In addition, as Mahrt (2017) stated, in very stable conditions, variation of turbulent fluxes on time-scales of a few minutes to tens of minutes are often associated with short periods of near-calm conditions where the turbulent fluxes are particularly small, and as a result its values are not very dependent on the averaging time.".

**6) I find difficult to follow the information given in Figs. 3-4, those where you show the wind roses. This is not the traditional way in which wind roses are represented (see for example Hullin et al., 2019; Fig. 2 for a better representation). By the way, I think this paper can be interesting for your present work and could be referenced. With regards to the information shown in these figures (3-4), I would like the authors to discuss more the differences found in wind direction distribution between Valimev and Skinflow towers, both for nIOPs and night-time whole dataset. For example, SE is clearly predominant for the Skinflow tower heights vs. Valimev for both datasets.**

Response: A different representation of wind roses is included in figures 3 and 4 in the new version of the paper. In addition, taking into account your comments and suggestions raised in questions 1, 2, 3 and 6, the Sect. 3 of the paper has been re-written.

As in the previous answer, this is included in several of the paragraphs of the new Sect. 3, between pages 6-9.

**7) I think it could be interesting to discuss how do you estimate the intermittency of the turbulence. I think it is not enough explained along the manuscript. Moreover, in the literature there are different definitions of turbulence intermittency, so it is important to know what you are using in the present study.**

Response: Considering your comment, we have added the following explanation along the text, in Sect. 2, page 5, between lines 10 and 17: "In the atmospheric boundary layer (ABL), intermittency is basically found on the stable boundary layer (SBL) over land at night, especially under conditions of large static stability and strong vertical wind shear, when turbulence is sporadic, characterized by bursts or episodes with periods of relatively weak or un-measurable small fluctuations. In this paper we use the term "intermittency", to describe specifically a temporal variation of turbulence intensity or turbulence strength, represented by the $V_{TKE}$ value calculated at a fixed location, as Sun et al., 2012. Reasons for increased turbulence are diverse and may include the intrusion of coherent structures such as gravity waves, density currents (Terradellas et al., 2005;

Udina et al., 2013; Ferreres et al., 2013; Soler et al., 2014) and low-level jets (LLJs) (Mahrt, 1999; Newsom and Banta, 2003; Cuxart and Jimenez, 2007).".

**Specific comments.**

**1) Revise the order of the references when you are citing more than one. Generally chronological order should be used, and this is not always done in the manuscript (see e.g. pg. 4, lines 13-14; pg. 7 line 9; pg. 14, line 5; …).**

Response: We have corrected for all the references in the wrong order along the manuscript (E.g. pg. 4 line 15)

**2) Pag. 2, lines 19-21: some reference could be given in relation with the TTE concept (Zilitinkevich et al., 2007, for example that you already have at the reference list).**

Response: We have added this reference at page 2, line 21.

**3) Pag. 3, line 29: replace pikes by peaks**

Response: We have replaced it. Pg 3, line 30.

**4) Pag. 3, line 30: I think it is less than 45 km.**

Response: When we refer to 45 km we are alluding to the higher mountains at the head of the valley. We make it clearer by differentiating the different distance to the main topographic features, i.e. the entrance to the valley, the Pic du Midi Massif, and, further up the valley, the highest mountains.
You can find the changes in page 3 between lines 27 and 33.

**5) Pag 4, line 10-11: I do not understand this sentence. Could you please revise it? I do not find any relationship with the phrase that comes next.**

Response: We have modified the paragraph to make it clearer, can be seen at page 4, line 13 and 14: "Two datasets with different sampling frequency were used from Skinflow mast levels between $z_{2m}$ and $z_{8m}$: thermocouples (1 Hz) and sonic anemometers (10 Hz) (Table 1)."

**6) From my point of view, the information given in pag. 5 (lines 8-16) is difficult to understand as it is, and I think that it is not necessary and could be discarded. Maybe you can reference the papers by Said et al., but not giving the detailed information that comes next. However, I missed some post-processing information of the sonic data. For example, the kind of rotation applied (double rotation, planar fit?).**

Response: We have taken the suggestion into account in page 6 lines 1-3: "We follow the corrections and filters indicated in Said et al. (2011, a, b) and De Coster et al. (2011) report, where the post-processing applied to the BLLAST data is explained in detail (e.g. applying a planar-fit rotation to the wind direction dataset)."

**7) Pag. 6, lines 16-17. The reference Román-Cascón et al. (2018) is Román-Cascón et al. (2019) and the complete reference at the Reference list is also wrong (for example the title of the Journal); below you have the correct one.**

Response: Thank you, it has been checked & changed in page 6 lines 23-24.

**8) Pag. 6, lines 28-29: when you say at the lower levels, indicate exactly the levels considered. And in line 30, the same for the higher levels.**

Response: It has been checked & changed in page 7, line 8 and line 11.

**9) Pag. 7, lines 1-9. In this context, it can be interesting reference the results found by Jiménez et al. (2019).**

Response: As in 8), it is checked & changed in page 7 line 14.

**10) Pag. 7, lines 5-7: you relate the occurrence of NW at higher levels with a SBL height below the Valimev tower and strong synoptic forcing. Have you ckeck this point? Have you estimate the SBL the SBL height? From my point of view, when strong synoptic forcing is present then the nocturnal ABL height should be weakly stably-stratified and the ABL height should be quite larger than the 60 m.**

Response: In the new manuscript, Sect. 3 has been extensively modified, from page 6 to 9.

**11) Pag. 7, lines 5-7: I cannot see in Fig. 2a the very small valley you mention at the south of the Skinflow tower. Could you give more information on this gully (slope and orientation)? It can be quite interesting to know it.**

Response**:** We have included an arrow pointing the gully in Figure 2.b (Page 4) where the gully can be seen at the south-east of the Skinflow mast. This small angle slope is orientated towards the NW.

**12) Pag. 7, lines 10-13: When you reference the SDF described by Román-Cascón et al. (2015), it is said that it ranges from noon 1$^{st}$ July to morning 2$^{nd}$ July, including nIOP8. This is wrong. The period analysed in Román-Cascón et al. (2015) ranges approx. from 1800 to 2200 UTC on the 2$^{nd}$ July (IOP10), and the SDF lasts from 1930 to 2030 UTC approx.**

Response: Thank you for correcting these details. We have changed the reference to the correct period in page 7 lines 14 and 15.

**13) Pag. 7, line 13: Change 1$^{st}$ July by 2$^{nd}$ July.**

Response: As in 12), we have modified it in page 7, line 17.

**14) Pag. 9, lines 1-2: It is said that the MPF is from SW. However, in pag. 6, line 31 it is mentioned that MPF comes from the SE quadrant. Could you explain this contradiction?**

Response: As this section has been extensively modified and improved, this contradiction is no longer in the text.

**15) Pag. 10, lines 30-32 and pag. 11, lines 1-2: A comment in the line of that done in comment 10); you justify that regime 2 does not behave as HOST for the 60 m for the largest winds because this height could be above the height of the SBL. However, I would not expect this just for the highest winds, where the NBL can be higher due to the mechanical turbulence generated for stronger winds.**

Response: We would like to remark that this phenomenon is mainly found due to the scarcity of measurements for these wind speed bins at these levels. Therefore, no real conclusions can be drawn from it. We have proceeded to write it with more emphasis in the paper, as can be seen in page 11 lines 9-12.

**16) Pag. 12, lines 13-15: "the turbulence intensity can be enhanced due to the presence of coherent structures". My question is: for what range of wind speed do you think it is more relevant the present of coherent structures (CS) and why? In relation with this question, in the same page, line 18-21, it would seem that you have more presence of coherent structures. Do you really think that CS are related to NBL more than SBL? Could you please clarify this in the discussion?**

Response: The presence of significant turbulence (intermittent burst) created by coherent structures is a characteristic of very stable boundary layers with light winds and large stability conditions (Terradellas et al., 2005; Udina et al., 2013; Ferreres et al., 2013; Soler et al., 2014). We introduced the coherent structures in the Introduction, page 2, line 11-14.

**17) Pag. 12, line 26: 'big difference'. Could you explain these differences? In line 28, when you use 'In addition', it seems that you are going to discuss about Skinflow tower, but you are referring to Valimev tower. Clarify it please.**

Response: This text has been removed in the new manuscript.

**18) Pag. 16, lines 6-7: 'There are few outliers in z3m and z5m since the surface smooths the quick shifts of wind speed and direction'; could you please explain better this sentence?**

Response: In the new manuscript, this phrase has been erased.

**19) Pag. 17, lines 3-8: in this paragraph you are discussing the presence of outliers in the SE's directions, and in part is related to the presence of storms and low pressure systems affecting this region. I think that at least low-pressure systems are related to NW's not SE's directions.**

Response: Fig. 8. and 9 refer (Sect. 5) to all the dataset, where SE's wind direction range not only are related to mesoscale flows, but can also include synoptic scale phenomena, as for example a low-pressure system. As can be seen in the BLLAST daily forecast report, for e.g. days 25 and 26 of June. The link to the webpage is: http://boc.sedoo.fr/source/dirCurrent.php?current=20110625&nav=Dailyforecastreport

**20) Pag. 19, line 5: when you mention 'atmospheric disturbances', at what scales are you referring to? Are internal gravity waves or other submeso motions important in this context?**

Response: The sentence is rewritten in page 18, lines 6-8, as: "Category B corresponds to the enhancement of turbulence within the regime 1 caused by atmospheric disturbances as internal gravity wave, density currents, and low-level jets (LLJs), that increases local turbulence and may reduce the local stability, even inducing some intermittency.".

**21) Pag. 19, lines 7-9: why don't you consider SW's instead of nIOPs to illustrate the intermittency categories? In fact, you can have suitable conditions (SBL) even when an IOP is not defined in BLLAST.**

Response: We have considered nIOPs because in BLLAST field campaign not all the SE's wind directions correspond to nIOPs and the intermittency categories are defined for very stable conditions.

**22) Pag. 20, lines 1-11. The paper from Román-Cascón et al. (2019) does not use BLLAST data (this is done in Román-Cascón et al., 2015), although they characterize the thermally-driven flows at the BLLAST site. So please cite both papers properly.**

Response: Thanks for your comment. See correction at page 19, lines 4-14 and page 20 line 1.

**23) Pag. 20, lines 10-11: 'The categories can also be found during other nIOPs'. Please indicate explicitly those nIOPs.**

Response: This has been included in page 19, line 13-14, in Sect. 6.1: "Category A can be found in all the nIOP cases but nIOP10. Category B is not so frequent, occurring for short periods during the nIOPs 01, 02, 03, 04, 05, 07, 09.".

**24) Pag. 21, lines 19-23: Can you indicate any references at the end of this paragraph?**

Response: We have included some references at page 20, line 25-26.

**25) I cannot find the purple line, corresponding to the 30m height.**

Response: As stated in the caption of former Fig. 12, now Fig. 11 (Sect. 6.1, page 21), the virtual potential temperature of the 30-m level is not displayed due to the limited availability of the Skinflow mast thermocouple data.

**26) Pag. 21, Figure 12 caption: I understand that category A is related to MP flow and category B to SDF, so it would be clearer if you state: Both stages… enhancement of turbulence… and transitions between reg. 1 to reg. 2 respectively.**

Response: We have modified the text so that it is clearer. Now this figure caption corresponds to the new Fig. 11 (Sect. 6.1, page 21) and the new text is as follows: "The

shallow drainage flow (SDF) stage is indicated between 1855 and 2020 UTC, and a consecutive mountain-plain flow (MPF) stage thereafter. In the SDF stage there are oscillations within regime 1, which is category B, whereas in MPF stage there are transitions between regime 1 and regime 2.".

**27) Pag. 24, line 12-13: 'mesoscale and synoptic scale meteorological situations'. According to this statement, mountain breezes, SE's are not a mesoscale flow?**

Response: Taking into account the 3$^{rd}$ general comment by the referee, mesoscale is replaced by synoptic scale in page 23, line 17.

**28) Pag. 24, lines 17: 'whole nocturnal dataset' or it should say 'whole SE's nocturnal dataset'?**

Response: As you noted, it should say 'whole nocturnal dataset'. We have corrected it in page 24, line 2.

**29) Pag. 25, line 6: after C you could add 'related to turbulence intermittency'.**

Response: We have added "related to turbulence intermittency" in the same line. Now page 24, line 10.

**30) Pag. 25, lines 9-10: Could you explain how local shear can be generated by internal gravity waves?**

Response: We have modified it in page 24 line 13-16 to: "Local shear can be generated by internal gravity waves of relatively small wind speed amplitude so that the wind speed is lower than the threshold value. As a result, there is an increase of turbulence within regime 1 (Sun et al., 2012), which is the category B turbulence transition. Mahrt (2010a) found that with very weak winds and strong stratification (regime 1), turbulence can appear under the presence of gravity waves.".

**31) Pag. 27, lines 33-35: This paper is already published, and the right journal is 'Atmospheric Research' and not 'Atmospheric Environment'. Please change it.**

Response: Thanks for the comment, the journal citation has been corrected in page 28 lines 9-11.

[revised manuscript text omitted]

---

## Author Comment (AC2) · 19 Jun 2019

Herein attached is the response to the referee #2 comments. We have appended the manuscript with all the changes on it. The struck through red lines represent the delated old text, whereas the blue text represents the new text that appears in the revised manuscript.

Please also note the supplement to this comment:
https://www.atmos-chem-phys-discuss.net/acp-2018-1300/acp-2018-1300-AC2-supplement.pdf

2019.

**Supplement:**

This document answers the questions on several aspects of the paper herein discussed. We would like to acknowledge the suggestions made by the referee.

**MAJOR SUGGESTIONS:**

**1. I do not think the entire section showing how the IOP cases behave is necessary. Very similar plots are shown in the following section, specially given that one of the wind directions considered coincides largely with the occurrence of IOP cases. It seems to me that no major conclusions are drawn from the IOP analysis alone. In fact, when reading that initial section of the paper, I was a bit disappointed, because I felt I was just reading an assessment on how the findings of Sun et al. (2012) apply to a different dataset. In the following section, on the other hand, there is novelty, related to the different hockey sticks and their dependence on the upwind terrain characteristics. Why not start with that, then? Moreover, why to keep the analysis on the IOP cases?**

Response: We realized that Sect. 4 could be shortened in order to make the manuscript easier to read and follow. The solution has been to move most of the content of the box plot analysis done in Sect. 4.2 to Sect 5.2, and the box plot figure, Fig. 6, to an Appendix. Now, Sect. 5.2 also includes the comparison between the box plots for nIOPs and the SE's for the whole nocturnal dataset, the discussion and the conclusions. The result is such that Sect 4 is shortened, and it does not include subsections.

However, we believe Sect. 4 is a very important part of the manuscript, and maybe this was not clear enough in the introduction and objectives. Thus, we have highlighted this importance in the objectives of the paper, as can be seen in page 3, lines 16-18. We have kept a part of it for several reasons:

1) We believe that this section is the basis for the analysis of all data examined in this study. In this way, when all data is considered, we already know that an important part of the SE wind directions correspond to nIOPs (between 63% and 70%), the remaining percentage corresponding to non nIOPs cases, which present important differences respect to the nIOPs dataset.
2) With reference to dropping Figure 3, we think that is important to keep it as it shows that the SE wind directions in the nIOPs cases correspond to the shallow drainage and mountain plain flows, completely linked to the topography of the area. This result will later allow us to easily interpret the results of Figure 4 when the analysis of all data is done. On the other hand, the analysis and comparison of Figures 3 and 4 has allowed us to perform the study of all the data in a coherent way (section 5), that is, we were able to perform a clear analysis separating the data by wind directions, so called NW's and SW's.
3) Finally, the study of the nIOPs cases has allowed us to isolate the influence of the topography on the stability regimes proposed by Sun et al., (2012) and to

verify its validity, which on the other hand was one of the objectives of this work.

2. **I would like to see some deeper development on the intermittency analysis. Maybe some statistics on how common each classes are, if they are favored under some type of external condition (large-scale wind direction) and how they affect the thermal structure.**

Response: We also think a deeper analysis on the intermittency categories would be interesting and useful. In fact, we believe this could be a very interesting further work. However, this is out of the scope of the present manuscript, as the main focus was to explore the HOST pattern in a complex terrain area. The turbulence intermittency examples are added in order to show specific cases of the nature of turbulence intermittency inherent within the HOST relationships, but it is not planned as a full statistical analysis of the BLLAST data. We hope you understand that we stopped at this point the analysis to avoid making the manuscript too long.

However, for the sake of clarity, in page 5 lines 9-17 we have added a deeper definition of the turbulence intermittency and the conditions under which it takes place.

**MINOR SUGGESTIONS**

3. **Page 5, lines 10-12. It is not necessary to tell the rotations you applied to the different sonic data. These are associated to the sonic deployment, which is specific to your experiment.**

Response: The text has been modified as can be seen in page 6 lines 1-4.

4. **I would drop Fig. 3, for reason detailed in point 1, above.**

Response: This question has been answered before, in the first major suggestion.

5. **Page 12, lines 7-12. This type of explanations belongs at the figure caption.**

Response: We have rephrased the first and the second paragraph of Sect. 5.2 so that in the first one (page 14 lines 12-14 and page 15 lines 1-3) there is a brief description of the box plot type considered and in the second one there is a discussion of the results (page 15 lines 4-9). Following your advice, the more technical information has been included in the figure caption (Fig. 8).

6. **Page 12, line 33. It is very difficult to understand what "too deep to calculate the bulk potential temperature difference" means. Usually it is easier to evaluate gradients over deeper layers, as they are larger.**

Response: We have modified this sentence because the text induced to misunderstanding. Now it is in page 11 lines 17-19 as follows: "We only proceed for the Skinflow mast because the separation of the measurements levels in the Valimev tower are too deep to calculate the stratification based on the bulk potential temperature difference, $\Delta\Theta_v$ (Sun et al., 2016)".

**7. Page 14, lines 8-10. This issue can be easily solved by using the thermal gradient dT/dz instead of temperature difference alone.**

[revised manuscript text omitted]